# Near-continuous observation of soil surface changes at single slopes with high spatial resolution via an automated SfM photogrammetric mapping approach

Oliver Grothum[1], Lea Epple[1], Anne Bienert[1], Xabier Blanch[2], Anette Eltner[1]

[1]Institute of Photogrammetry and Remote Sensing, TUD Dresden University of Technology, Helmholtzstr. 10, 01069 Dresden, Germany

[2]Department of Civil and Environmental Engineering, Universitat Politècnica de Catalunya, 08034 Barcelona, Spain

*Correspondence to:* Oliver Grothum (oliver.grothum1@tu-dresden.de), Anette Eltner (anette.eltner@tu-dresden.de)

**Abstract.** Soil erosion represents a major global threat, necessitating a detailed understanding of its spatial and temporal dynamics. Advanced geospatial technologies such as time-lapse structure-from-motion (SfM) photogrammetry provide high-resolution monitoring of surface changes. This study presents a novel event-driven approach for near-continuous monitoring of hillslope surface dynamics over a multi-annual period. The system employed synchronized digital single-lens reflex cameras at three slope stations, triggered by a rain gauge and a daily timer. Ground control points (GCPs) were surveyed with millimeter accuracy to ensure precise georeferencing.

An automated Python-based workflow was developed to synchronize images, detect GCPs using a convolutional neural network, generate daily digital 3D surface models via SfM, and compute 3D surface models of difference. The absolute accuracy of SfM point clouds ranged between 8 mm and 12 mm on average, primarily due to registration errors, with lower deviations (< 5 mm) in central areas after height adjustment. Relative accuracy decreased concentrically with distance from the cameras, with level of detection values between 5 mm and 25 mm depending on distance and location.

Time series analysis revealed surface changes driven by rainfall, snowmelt, and agricultural activity. The most significant changes often occurred shortly after tillage, even with minimal rainfall, indicating both erosional and non-erosional processes. A strong negative correlation between rainfall and elevation loss was especially evident within the first seven days following tillage. Seasonal surface lowering of 3-5 cm during winter and occasional positive changes due to frost or vegetation growth were also observed. The monitoring system and workflow are transferable, and the resulting high-resolution datasets are expected to be valuable for analyzing erosion dynamics and testing process-oriented soil erosion models.

## 1 Introduction

Soil erosion represents a significant global threat to both soil security and human security, contributing to land degradation and compromising vital ecosystem functions (Doetterl et al., 2016, Borrelli et al., 2017). Effective land management and soil conservation require a detailed understanding and quantification of erosion processes across spatial and temporal scales (e.g., Jetten & Favis-Mortlock, 2006, Fiener et al., 2020). A variety of soil erosion models have been developed to assess erosion potential and support mitigation strategies, ranging from empirical approaches such as the Universal Soil Loss Equation (USLE) and its revised form (RUSLE) to process-oriented models like the Water Erosion Prediction Project (WEPP) (e.g., Karydas et al., 2014, Panagos et al., 2015, Batista et al., 2019, Epple et al., 2022). While the application of these models is well established, there remains a need for harmonizing methodologies and improving model evaluation strategies (e.g., Kohrell et al., 2023, Eltner et al., 2025), which critically depend on the quality and resolution of the input data (Doetterl et al., 2016, Batista et al., 2019, Fiener et al., 2020).

The accuracy of such models is closely tied to how well the surface and topographic conditions are characterized. Traditional methods, such as rainfall simulation experiments and sediment yield measurements, have been augmented in recent years by advanced geospatial technologies (e.g., Hänsel et al., 2016, Balaguer-Puig et al., 2017, Li et al., 2023). High-resolution surface data obtained through structure-from-motion (SfM) photogrammetry and terrestrial laser scanning (TLS) offer promising avenues for capturing surface dynamics associated with processes such as interrill and rill erosion on hillslopes at the event scale (e.g., Eltner et al., 2015, Cândido et al., 2020, He et al., 2022, Dai et al., 2022, Evans et al., 2024). These methods enable the detection of subtle changes in surface microtopography and provide spatially continuous information essential for model calibration and evaluation (Epple et al., 2022).

The generation and processing of three dimensional (3D) point clouds for observing and understanding earth surface processes has become a cornerstone of geomorphological research. Early systems for change detection relied on simple camera setups combined with external depth data (e.g., Krimmel & Rasmussen, 1986; Schwalbe et al., 2016). With the advent of SfM photogrammetry, studies began constructing time series of 3D surface data directly from image sequences, often collected through manually triggered, synchronized DSLR (digital single-lens reflex) setups (Eltner et al., 2017) or automated low-cost systems using microcontrollers and wireless data transmission (Kromer et al., 2019; Blanch et al., 2023, 2024). Additionally, laser scanners mounted on permanent structures have enabled continuous monitoring of highly dynamic environments such as beaches (Vos et al., 2022), supporting the development of advanced four dimensional (4D) change detection algorithms (Anders et al., 2021).

In the context of soil erosion research, this technological progress enhances our ability to observe and analyze processes such as splash, interrill, and rill erosion with unprecedented detail. Artificial rainfall simulations remain a key approach for producing controlled erosion data on standardized plots (Michael, 2014, Hänsel et al., 2016), which serve as benchmarks for calibrating and testing process-oriented soil erosion models (Schindewolf & Schmidt, 2012). However, extending these observations to the field scale is critical for upscaling model applications (Eltner et al., 2018), as it captures the spatial heterogeneity and complexity of real-world erosion dynamics.

In this study, we present a novel, event-triggered camera system developed for near-continuous monitoring of soil surface changes on an experimental field on a hillslope over a period of 3.5 years. The system uses synchronized cameras and a partly open-source workflow to automatically organize image data, detect ground control points (GCPs), and reconstruct daily 3D surface models (i.e., 3D point clouds). These models are assessed using reference data from TLS and uncrewed aerial vehicle (UAV) photogrammetry to measure absolute accuracy, while relative accuracy and the level of detection (LoD) are estimated from multiple 3D surface models acquired under unchanged surface conditions. This approach enables high spatial and temporal resolution observations of soil surface changes caused by natural rainfall, snowmelt, and agricultural practices, capturing a range of erosive processes and supporting multi-scale model development and testing.

## 2 Experimental design

To quantify and monitor soil surface change dynamics during natural rainfall events, we propose a semi-autonomous observational system that integrates event-triggered camera control with time-lapse photogrammetry.

### 2.1 System setup

A monitoring system was installed on freshly tilled grassland at an agricultural test site in the hilly loess region of Saxony, Germany, to record soil surface changes at the field scale almost continuously over an extended period. The field was kept

free of vegetation by frequently grubbering the soil at depths of 5-20 cm. The experimental site has a north-northeast orientation, and the slope reaches a steepness of up to 14%. The soil is classified as sandy silt with a soil organic carbon content of 1.25% (as of July 2021), making it prone to erosion.

The hillslope was observed over a length of about 60 m and a width of about 15 m. Three monitoring systems were installed: one at the top, one in the middle, and one at the bottom of the slope. Each observation system consisted of two A-shaped constructions carrying a wooden horizontal beam (Fig. 1). On these cross struts, five DSLR cameras were mounted, which triggered by microcontrollers. The original system consisted of one metal traverse and two angular elements to maintain horizontal stability. However, during a storm, two of the monitoring systems collapsed, prompting a redesign of the weaker construction elements. The traverses were subsequently supported by additional horizontal planks and two stronger angular wooden elements installed below them. Furthermore, the system was installed such that a natural, vegetated boundary along the periphery of the study area was maintained. While the absence of a closed hydraulic boundary means that our measurements do not represent sediment yield from a hydrologically isolated system, they can be directly used to quantify volumetric soil erosion within the defined area, measured as net change in surface elevation.

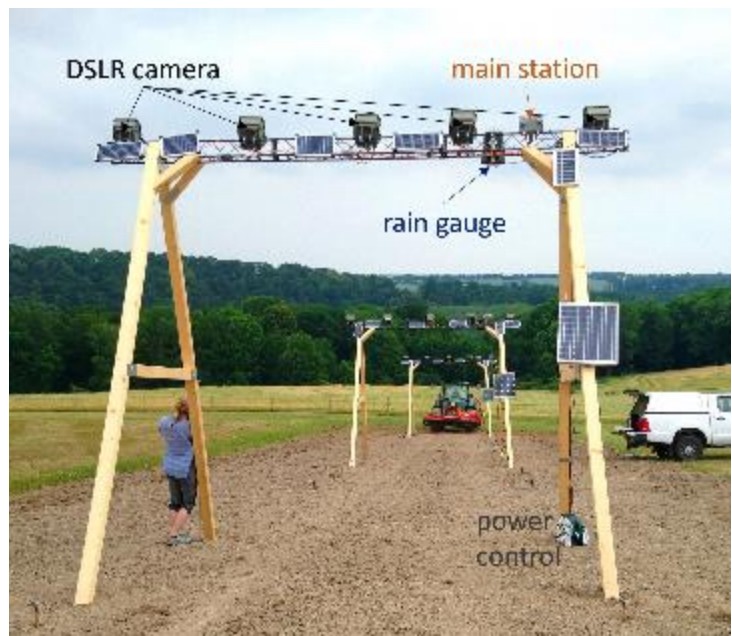

**Figure 1: Construction of the soil surface monitoring system at the lower, middle, and upper slope positions observing the hillslope field with synchronized DSLR (digital single-lens reflex) cameras that were triggered by a rain gauge. The image shows the regular upslope and downslope tillage of the field.**

### 2.2 Data acquisition

Each camera on the rigs was placed in a waterproof housing. Power was supplied by an external 7 Ah sealed lead-acid battery, which was charged by a 10 W solar panel placed near the camera housing. The triggering of the cameras occurred in two ways, controlled by two microcontrollers: (a) a scheduled trigger and (b) an event-based trigger due to rainfall. In the first case (a), a permanently running Arduino Micro Pro, connected to a real time clock (RTC), sent a trigger signal every day at 10:00 a.m. CET (Fig. 2). The microcontroller was powered by the same battery-solar panel system as the cameras.

The second, event-based trigger (b) originated from the main station, which was controlled by an Arduino MKR FOX 1200 (Fig. 3) connected to an RTC module. The main station controlled the synchronized camera trigger. Its microcontroller was coupled to a rain gauge consisting of a reed switch, which was activated every time 0.2 mm of was collected by the tipping bucket. This closed a circuit that was registered by the Arduino. Each time a trigger signal was received, the main station microcontroller switched the state of a relay that was connected in parallel to all five cameras, allowing synchronized image

capture. In addition to the rain-controlled trigger, the main station also initiated a synchronized daily image capture at 10:00 a.m. CET. Besides the trigger connections, the main controller was connected to additional weather sensors: a DTH11 sensor to measure air temperature and humidity, and a soil moisture sensor. The sensor readings were transmitted via IoT (internet of things) using the infrastructure provided by the company Sigfox. which operates a low power wide area network. To conserve energy, the microcontroller was set to deep sleep mode and woke only upon the daily alarm interruption signal (10:00 a.m. CET) or an external interrupt signal triggered by the rain gauge bucket. The main station was powered by a 50 Ah sealed lead-acid car battery, charged by a 70 W solar module.

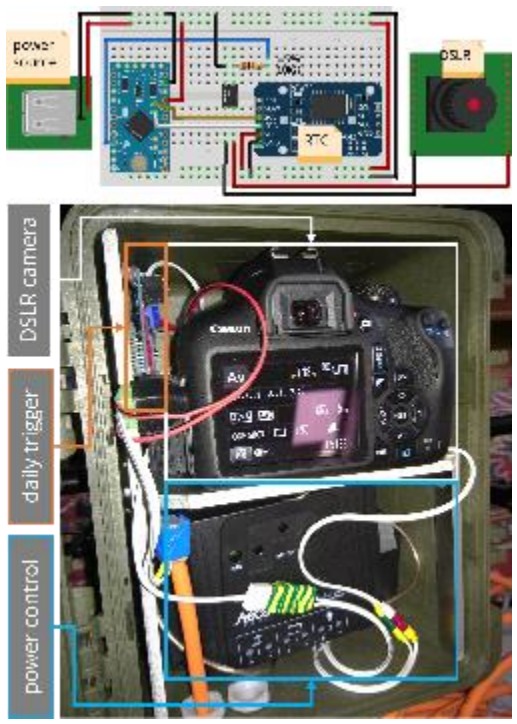

**Figure 2: Housing and internal components of one of the five cameras installed at each monitoring rig. The top sketch (created with fritzing.org) illustrates the wiring of the daily trigger system using an Arduino (DSLR = digital single-lens reflex, RTC = real time clock).**

The DSLR cameras were Canon EOS D2000 models, each equipped with a zoom lens fixed at 18 mm using tape. The cameras had a 3.5 mm 3-pin jack interface for external triggering. They were set to aperture priority mode to ensure geometric consistency of the internal camera orientation throughout the observation period. Exposure time was left variable to allow flexibility under different lighting conditions.

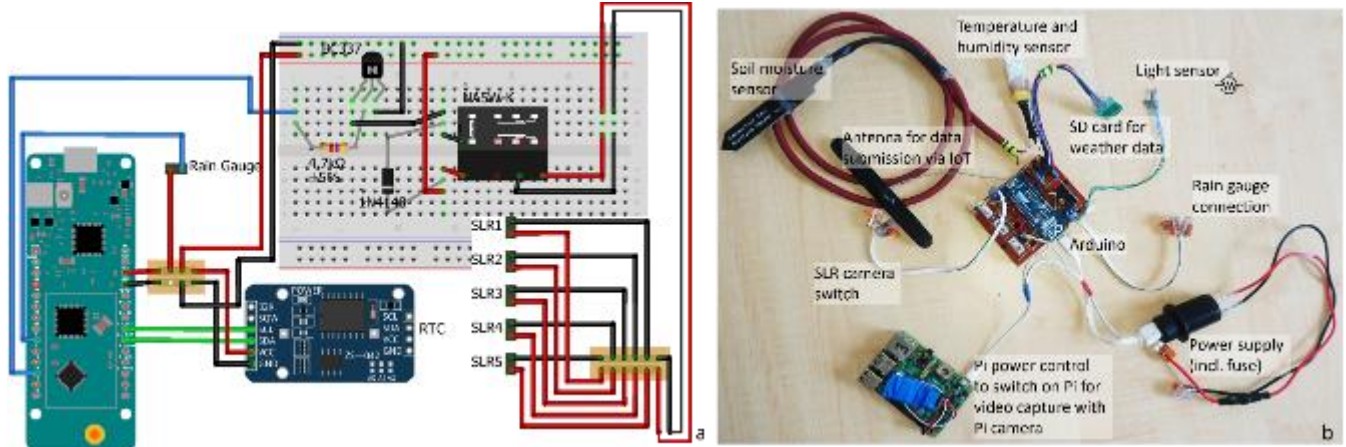

**Figure 3: (a) Schematic of the simplified parallel camera triggering system (main station created with fritzing.org), shown without environmental sensors and Raspberry Pi . The system uses a rain gauge for rainfall measurement and camera triggering, and a RTC (real time clock) for synchronous triggering at 10:00 a.m. CET. (b) Setup used in the field, including all sensors and systems (SLR = single-lens reflex).**

### 2.3 Georeferencing

GCPs were installed in the field to orient the image measurements within a reference frame and to scale the photogrammetrically derived 3D models (Fig. 4). The GCPs consisted of white circles on a black background. The markers were attached to poles driven up to one meter deep into the soil to ensure stability during the monitoring period, an important requirement for soil surface change detection (Eltner et al., 2015). The GCPs were slightly tilted toward the camera rigs to maintain visibility and to minimize distortions from the bird's-eye perspective in the case of UAV data. The coordinates of the GCPs were measured using a Leica TCRM 1102 total station, achieving millimeter-level accuracy. Marker position estimates were transformed from polar to Cartesian coordinates based on the total station's orientation, resulting in a final coordinate for each point with an accuracy of 2 mm.

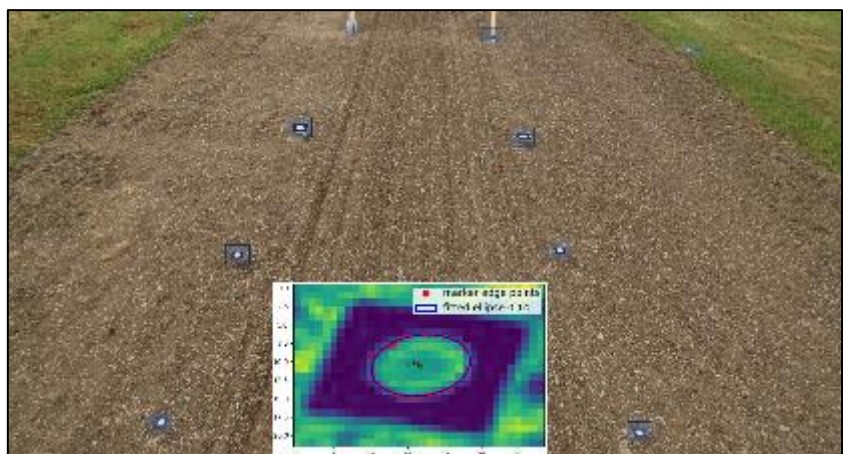

**Figure 4: GCP configuration at one camera rig. The ten markers outline a trapezoidal area of interest. The inset shows the ellipse fitting result for one GCP.**

### 2.4 Reference data for the accuracy assessment

#### 2.4.1 UAV photogrammetry

A DJI Phantom 4 RTK drone was used to capture the UAV-based imagery on 2020-08-11. Due to the low flight altitude of 10-15 m, required to achieve millimeter-level image resolutions, the flights and image triggering had to be performed manually. The flight pattern was designed to cover the entire area of interest with nadir images (i.e., camera gimbal angle of 90°) and convergent images taken from the sides of the field at 60° angles. The images were processed in *Agisoft Metashape* (v1.8.3), using the standard SfM photogrammetry pipeline (e.g., Eltner & Sofia, 2020), which involved establishing the relative orientation of all images, referencing them to the local coordinate system defined by the GCPs, and constructing a dense point cloud of the field surface.

#### 2.4.2 Terrestrial laser scanning

To check the processed camera data for systematic errors, the soil surface was scanned with a RIEGL VZ400i terrestrial laser scanning system. The investigated slope was scanned on 2020-08-11. According to the manufacturer's specifications, the device can capture half a million points per second with an accuracy of 5 mm at ranges of up to 800 m. Scanning was performed from eight different positions surrounding the hillslope. Each scan was relatively oriented to its neighboring scan using the iterative closest point (ICP) algorithm (Besl & McKay, 1992; Chen & Medioni, 1991), minimizing the mean distances between point clouds.

To transform the laser scanner point clouds into the local coordinate system, retro-reflective cylindrical targets were used to identify reference points during scanning. The cylinders were placed on the GCPs (used for the image measurements) that bounded the field. By matching the targets detected in the scanner coordinate system to the GCP points in the local system,

the parameters of a rigid transformation were calculated. Applying the estimated translation and rotation parameters enabled the transformation of the scanner point cloud into the local coordinate system.

## 3 Methods

The automatic workflow for time lapse-based soil surface change detection is divided into five modules: (a) camera calibration, (b) image sorting and time attribution, (c) time lapse SfM photogrammetry, (d) point cloud processing for filtering the reconstructed 3D surface models, and (e) calculation of the digital elevation model of difference (DoD). The implementation is realized in Python. The workflow primarily utilizes the *NumPy*, *pandas*, *matplotlib* and *OpenCV* libraries for data management, visualization, and image processing. For image alignment and 3D reconstruction of the soil surfaces, the *Agisoft Metashape Python API* (v1.8.3) was used. Point cloud processing steps were performed using the Python package *CloudComPy*, which provides access to various *CloudCompare* (v2.13.2, 2024) functionalities.

### 3.1 Camera calibration

Calibrating cameras is a fundamental task for accurately modeling the projection ray from the object point to the image sensor plane. In this study, the camera parameters were modeled using the approach of the Brown (1971), which includes the focal length ($f$) and the principal point coordinates ($xP$, $yP$), representing the perpendicular intersection of the focal point with the sensor plane. In addition, image distortion was accounted for through radial distortion parameters ($K1, K2, K3, K4$) and decentering distortion parameters ($B1, B2$).

Each camera was pre-calibrated, since only a limited number of images (i.e., five) were available for the 3D reconstruction, and a different set of parameter values was required for the interior orientation of each camera. This approach contrasts with typical SfM applications, where many images are taken with a single camera. The aim was to avoid over-parameterization by estimating distortion parameters beforehand and deriving reliable approximations of focal length and principal point, which were later only fine-tuned during the reconstruction of the area of interest (Eltner et la., 2017).

To estimate the interior geometry parameters, images of a calibration field with markers of known geometry were captured and processed in a least-squares bundle adjustment to derive both the interior and exterior orientations (Luhmann et al., 2016). During pre-calibration, approximately 50 coded markers were arranged in a square pattern to serve as a temporary target field (e.g., Eltner & Schneider, 2015). To decorrelate the focal length and camera-to-object distance estimations, additional markers were placed on small boxes to introduce height variations in the calibration field (Fig. 5). Marker coordinates were initially measured with a measuring tape (to millimeter-level accuracy) to define a local reference frame. These coordinates were refined during the bundle adjustment and were not treated as fixed control points, since they were not measured with superior accuracy and were therefore considered as approximate values. Images were taken by each camera while moving around the calibration field, following the acquisition scheme proposed by Godding & Luhmann (1992) to minimize parameter correlations. Calibration was performed using *Aicon* (version 1.12). As *Aicon*'s calibration outputs are not directly compatible with *Agisoft Metashape*, the calibration images, object-space coordinates, and image measurements of the coded targets were exported from Aicon and imported into *Agisoft Metashape*. The coded target coordinates were introduced as GCPs with assigned accuracies of 0.1 mm to constrain them during parameter estimation. The resulting calibration parameters were saved and later applied in the time lapse SfM processing step.

Camera calibration was performed once prior to the initial system setup and repeated after the rig collapse and subsequent reinstallation. No further calibration was required, since only approximate values for the focal length and principal point were needed; these parameters were re-estimated during each model calculation. Distortion parameters were estimated once and assumed to remain temporally stable throughout the monitoring period.

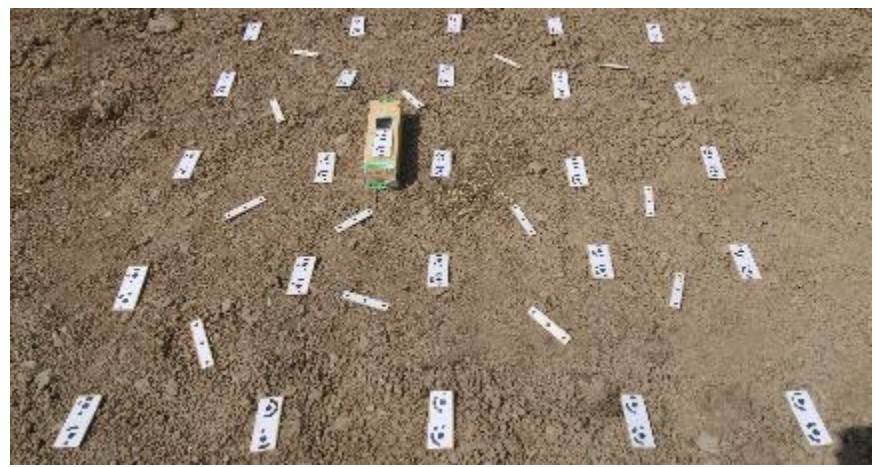

**Figure 5: Control point scheme for camera calibration adjacent to the experimental field.**

## 3.2 Time attribution

Synchronizing multiple cameras for time lapse data collection presents a particular challenge due to temporal drift in both internal camera clocks and external triggers. Such drift can accumulate to time discrepancies of up to 30 minutes over several months (Mallalieu et al., 2017). In addition to drift, further inconsistencies in the image series can arise, for example when cameras fail to capture an image despite receiving a trigger signal. To address these issues, an automated alignment method was developed (Fig. 6).

| Epochs | Ref. camera | Camera 1 | Camera 2 | Camera 3 | Camera n |
|---|---|---|---|---|---|
| All | ▤ | ▤ | ▤ | ▤ | ▤ |
| Epoch 1 | ▢ | ▢ | ▢ | ▢ | ▢ |
| Epoch 2 | ▢ | ▢ | ▢ | ▢ | ▢ |
| Epoch 3 | ▢ | | ▢ | ▢ | ▢ |
| Epoch 4 | ▢ | ▢ | | | ▢ |
| Epoch 5 | ▢ | ▢ | ▢ | ▢ | ▢ |
| Epoch m | ▢ | ▢ | ▢ | ▢ | ▢ |

**Figure 6: Temporal alignment of camera image series accounting for real time clock (RTC) drift and occasional capture failures during the long-term observation. Remaining cameras are matched to the chosen reference.**

The core of the process involves aligning image time series from multiple cameras. First, a reference camera is selected. Within monthly batches of images, the camera with the highest number of captures is chosen as the reference. To calculate reliable offsets, a minimum of one month of data is required; in cases of outages, a longer period (e.g., three months) is used. Accordingly, throughout the experiment, reference camera selection, offset calculation, and time attribution were performed in rolling three-month batches. The time offset of each camera relative to the reference is calculated based on daily images captured at 10:00 a.m. CET, triggered by both the main station and the Arduino Micro Pro backup system. A linear model is fitted to the time series data from each trigger source, and the source with the lower root mean square error (RMSE) is selected to define the offset. This procedure creates a lookup table for future time alignment. The calculated offset is then applied to the image timestamps. Typically, the time series from the main station trigger is used for correction because it exhibits a more

consistent temporal offset. A 3-sigma rule is applied to ensure accurate matching with a high level of confidence. Thus, the absolute time difference between the reference image and its corresponding offset-corrected image must be less than three times the RMSE of the fitted line. If this threshold is exceeded, the temporal match is considered a failure (Fig. 7). Finally, images are assigned to their respective trigger events (i.e., first or second daily image) based on capture timestamps, with the earlier image captured belonging to the first series. The assumption is that the series maintain distinct, non-crossing offsets within the monthly to three-monthly period, which is visually confirmed by the variance of their capture times relative to the fitted offset line (Fig. 7). The series with the lowest variance (represented by the blue line in Fig. 7) is considered to originate from the more accurate main station trigger. Time offsets between the reference and the remaining cameras were determined monthly. Note, the reference camera can vary from month to month.

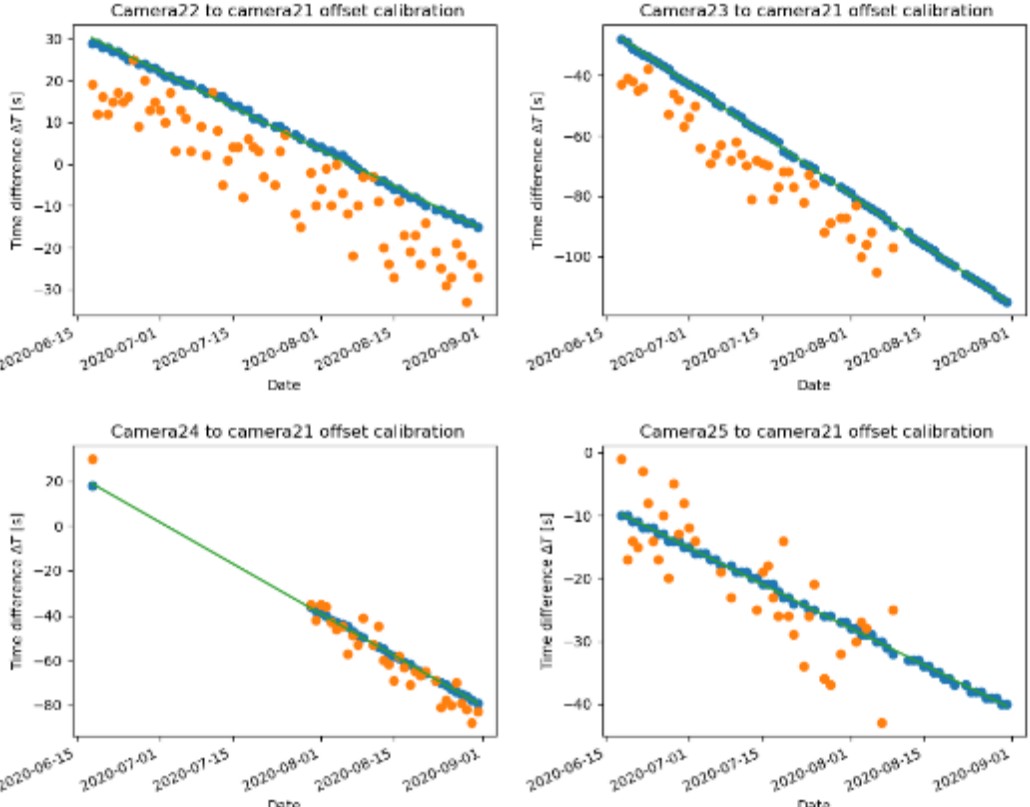

**Figure 7: Time offset calibration for each secondary camera relative to the selected reference camera for June, July, and August 2020. Blue dots refer to image capture times triggered induced by the main station, while orange dots represent images triggered by individual backup systems. The green line shows the least squares adjusted line of the blue, main station triggered data.**

### 3.3 Ground control point tracking

To reliably track GCPs under varying environmental conditions, a two-stage approach was developed. Initially, a custom Keypoint-CNN (convolutional neuronal network) was trained to detect GCPs more robustly than traditional template matching methods. This model uses a bounding box to locate potential GCPs and a keypoint detector to pinpoint their center coordinates. To ensure no GCPs were missed, images were processed in overlapping 512 x 512-pixel patches. Next a k-means clustering algorithm was applied to group multiple detections of the same GCP, with the final coordinate determined by averaging the cluster. For additional technical details, refer to Blanch et al. (2025).

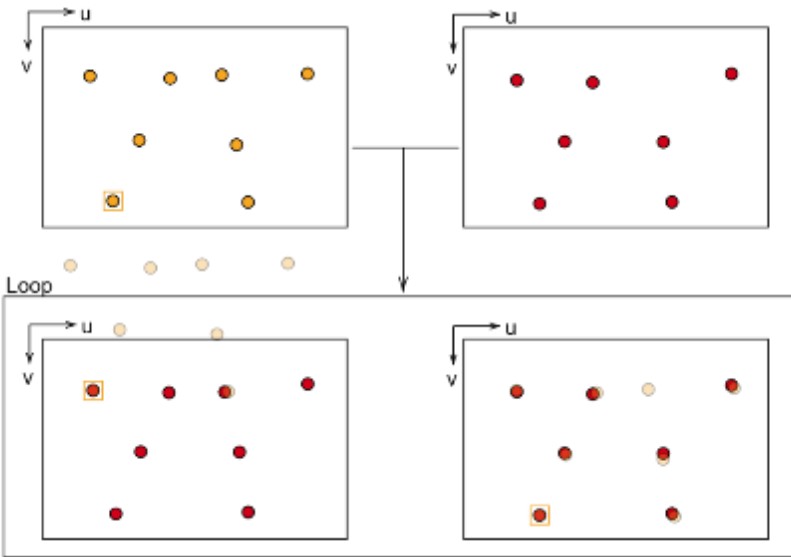

**Figure 8: Principle of the GCP labeling process. Orange dots represent the reference positions of labeled GCPs, while red dots indicate the detected GCPs without labels in subsequent images. The orange bounded GCP is the query point for a single iteration of the process. The bottom images show the loop, where reference points are shifted onto each detected GCP in the target image. The left image inside the loop shows a poor overlap and is rejected, whereas the right image shows good congruency, and the label of the reference query point is applied to the target GCP. u and v refer to the image coordinate system axes.**

Since the tracking algorithm does not assign unique identifiers, a labeling method was developed to match detected GCPs with their corresponding 3D coordinates. For label assignment, the first image from each camera each month serves as a reference image, where GCPs are manually measured and labeled (Fig. 8). These labels are then propagated to subsequent images using a spatial congruence algorithm that considers the relative spatial position of surrounding GCPs. The algorithm iterates over each GCP in the reference frame. The coordinates of all GCPs are reduced by the image coordinates of the query GCP. The reduced coordinates are then projected into the image frame of subsequent images. The center of the projected net is moved onto each locally detected GCP. If the projected net is congruent with the GCP image positions of the target image, the projected GCP and the measured GCP should align within a small error. The constellation with the most overlapping measurements within a threshold of 50 pixels is accepted. If fewer than four matches are found, the GCP is considered unreliable. This procedure is repeated for every GCP in the reference image, which is then treated as query point. To achieve sub-pixel accuracy, an ellipse fitting algorithm refines the detected the detected GCP coordinates. Any remaining errors in tracking or ellipse fitting were corrected manually. For independent verification of the 3D reconstruction accuracy, at least one GCP was designated as a check point (CP), i.e., it was excluded from the bundle adjustment process.

### 3.4 Soil Reconstruction

### 3.4.1 SfM photogrammetry

In this study, SfM photogrammetry was used for digital soil surface reconstruction. The process begins with the detection of key points, which are distinct features found in multiple overlapping images. These key points are matched by comparing multi-dimensional vectors that encapsulate the local image neighborhood. Matched features, also called tie points, serve as the primary observation for image alignment. The image observations are used in a bundle adjustment to estimate the parameters of the image network geometry. This adjustment simultaneously refines the internal camera parameters, camera poses (positions and orientations), and the 3D coordinates of the tie points through triangulation across multiple views. Georeferencing is achieved by considering the GCPs' image and object coordinates as additional constraints in the bundle block adjustment. With the established image network geometry, a dense point cloud can be calculated. This point cloud offers a highly accurate 3D representation of the soil surface, with a 3D coordinate estimated for nearly every pixel.

### 3.4.2 Time lapse data processing

The time lapse processing workflow is schematically illustrated in Fig. 9. The 3D reconstruction step begins by associating each image with its respective interior calibration parameters and organizing the images chronologically using the previously computed attribution table. The image sets are then imported into *Agisoft Metashape* and organized into chunks representing distinct reconstruction units.

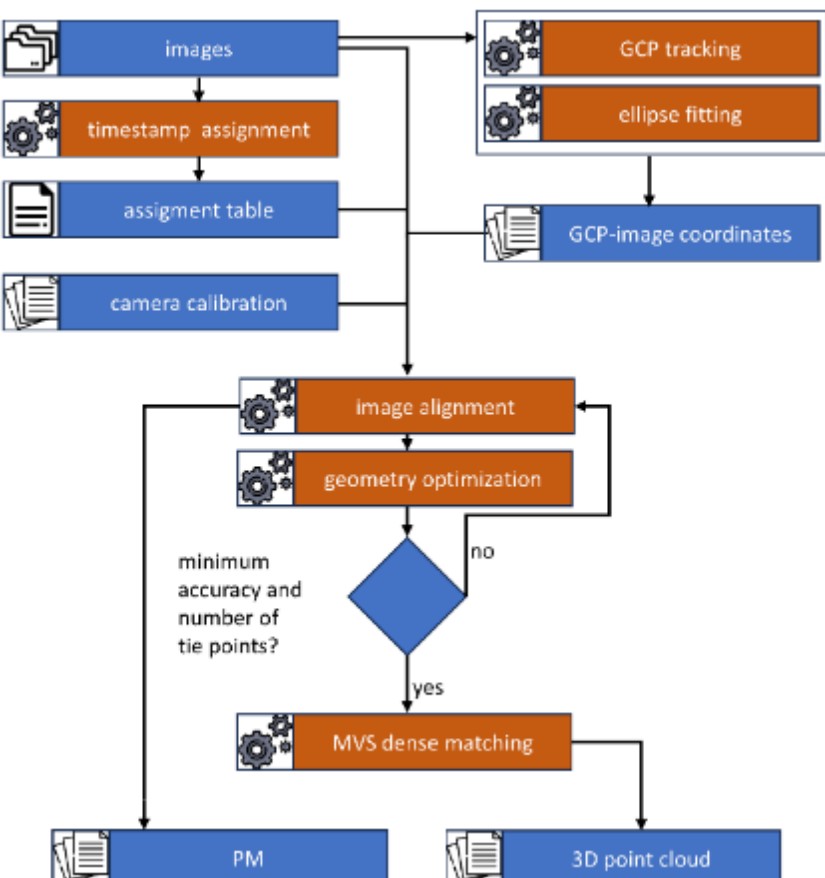

**Figure 9: SfM photogrammetry workflow for time lapse data processing (GCP = ground control point, MVS = multi view stereo, PM = precision map).**

Next, the tracked GCP locations are imported, and the image alignment is performed relative to the local coordinate system. During this step, the interior camera parameters are optimized, although only the focal length and principal point are adjusted. The number of tie points and the quality of the image measurements in pixel space are assessed to potentially refine the

alignment. If insufficient tie points are found (minimum required: 75) or the pixel error exceeds a given threshold (maximum allowed: 1.0 pixel), an iterative refinement process is applied:

1. Enable guided image matching – This allows camera calibration and initial rough alignment to improve tie point detection, increasing the number of point correspondences, though at the cost of longer processing times.

2. Upscaling image resolutions for alignment – If the initial alignment (performed with downscaled images by a factor

of two for speed) is inadequate, the original image resolution is restored.

3. Increase key-point and tie-point limits.

4. Filter tie points for stability and quality – The following criteria are applied: reprojection error (threshold: 1.0 pixel), reconstruction uncertainty (threshold: 25), projection accuracy (threshold: 4). Please, refer to the *Agisoft Metashape* user manual for detailed information (Agisoft LLC, 2023).

Filtered tie points are exported along with the standard deviations of their reprojections. These values are then used to estimate the 3D accuracy of the sparse point cloud following the method of James et al. (2017), forming the basis for the precision map

(PM), which spatially represents the LoD in the multi-temporal analysis. Finally, a dense point cloud is generated using *Metashape*'s high quality setting, which downscales images by a factor of two.

### 3.4.3 Point cloud processing

To compute the final 3D surface model for each reconstructed epoch, the dense point clouds and their corresponding PMs must first be filtered, combined, and pre-processed before they can be used for the calculation of DoDs (Fig. 10).

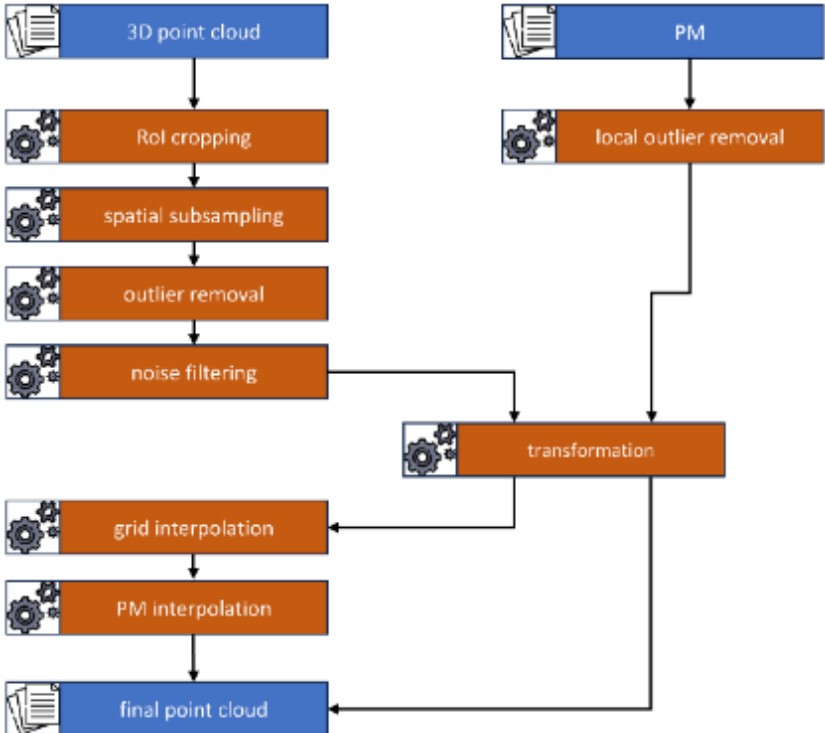

**Figure 10: Point cloud processing workflow (PM = precision map, RoI = region of interest).**

In the first step, all dense point clouds are cropped to the region of interest (RoI), approximately 100 m² in size. This is achieved
by projecting the point cloud onto the XY-plane and retaining only the points within a defined bounding box. Afterwards, the point cloud is spatially subsampled to reduce density, ensuring a minimum spacing of 2 mm between points. This step reduces dataset size and filters densely populated areas more aggressively than sparser ones, while still preserving fine-scale soil surface structures.

To mitigate systematic reconstruction errors, which appear as floating point clusters significantly above or below the actual
reconstructed soil surface, a best-fit plane is computed through the entire point cloud, and each point's signed distance to the plane is calculated. Points exceeding three times the standard deviation (3σ) of all distances are removed. To further filter remaining single outliers or small clusters of outliers, a local noise filter is applied: points are removed if the distance to the best-fit plane through their 30 nearest neighbors (NN) exceeds 2σ. Additionally, any point with fewer than 30 neighboring points within a 1 m radius is considered isolated point and is removed.

After the cleaning process, the point clouds are rotated parallel to the XY-plane, enabling later height-based comparisons parallel to the Z-axis. A plane is fitted only through the first point cloud of the series, and a transformation matrix is calculated by transforming it's normal vector to the Z-axis unity vector $ez = (0,0,1)$, following the algorithm proposed in Möller & Hughes (1999). The resulting transformation matrix is applied to the point cloud itself, its associated PM, and to all following 3D point clouds and PMs. The necessity of transforming the PMs is illustrated exemplary for a single point in Figure 11. Finally, the
cleaned and transformed point cloud is interpolated onto a uniform grid with a 5 mm resolution. This resolution was selected based on the camera-to-subject distance, representing an optimal balance between achieving the highest possible level of detail

and avoiding the introduction of interpolation artifacts caused by data gaps, which would have appeared at finer resolutions. Resolutions lower than 5 mm were not considered, as they resulted in an unacceptable loss of critical surface details while offering no improvement in terms of data quality or completeness.


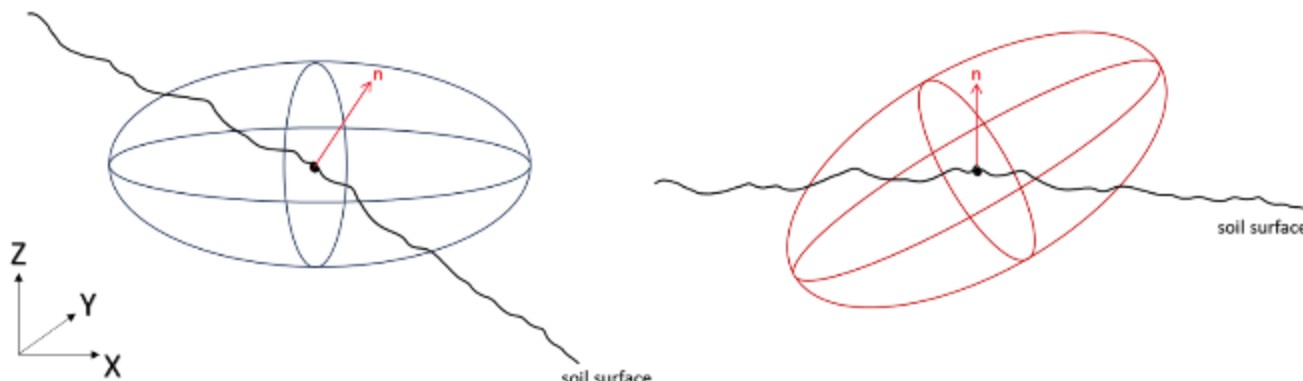

**Figure 11: Applying the transformation to the point cloud itself also requires applying the same transformation to the PMs (precision maps). Considering the precision values of a given point as a triaxial ellipsoid, the left image shows the situation in the local reference system, while the red vector shows the normal (n) direction and direction of the M3C2 (multiscale model to model**
**cloud comparison). By rotation of the surface into the XY-plane, the ellipsoid must be rotated accordingly to preserve the orientation of the normal vector within it.**

The dense point clouds and PMs are combined by interpolating the accuracy values from the PMs onto the dense point cloud. For each point of the dense cloud, the nearest PM points within a 1 m search radius are identified. The precision values are then calculated using a normal distribution kernel (Eq. 1).

$$f(d) = \frac{1}{\sigma\sqrt{2\pi}} e^{-\frac{1}{2}\left(\frac{d}{\sigma}\right)^2} \qquad\qquad 1$$

Defining a $\sigma = 0.3$ in this study, the weight of each PM point is determined by the distance $d$ of the query point to the PM point.

### 3.4.4 Change detection

The comparison of point clouds is performed using an adapted version of the multiscale model to model cloud comparison
(M3C2) algorithm (Lague et al., 2013), referred to as M3C2-PM (James et. al., 2017). In this method, a reference epoch is defined – usually the first epoch after the most recent tillage or camera system update – and all subsequent epochs are compared to this reference. In the M3C2 approach, the point normal $n$ is calculated for a core point $i$ using a fitted plane estimated from the neighbors $NN_i$ to the point. A cylinder with a radius $r = 50$ mm is created, oriented parallel to the core point normal $n_i$ and with its origin at $j$. The cylinder intersects both point clouds, extracting subsets of points from each. Each subset is then
projected onto the cylinder axis to estimate its average height value.

To account for the uncertainty of the mean distance estimation, the LoD at 95% confidence ($LoD_{95\%}$) is calculated using the propagated error values from the PMs. Since PMs are the result of error propagation of interior and exterior camera orientation during bundle adjustments, they allow for spatially correlated errors (James et al., 2017). The uncertainty is quantified by Equation 2.

$$LoD_{95\%}(r) = \pm 1.96(\sqrt{\sigma_{N1} + \sigma_{N2}} + reg) \qquad\qquad 2$$

Where $\sigma_{N1}$ and $\sigma_{N2}$ are the standard deviations of the averaged height value in normal direction for the two point clouds and *reg* indicates the data registration error. Eventually, significant changes for the point-wise differences will be indicated in a binary map, i.e., whether a statistically significant surface change occurred or not.

### 3.5 Accuracy assessment

The performance of the setup and data processing workflow was evaluated by comparing the reference 3D surface models obtained from TLS and UAV with the time lapse data of the same day, providing absolute accuracy estimates. All point clouds were leveled to the XY-plane to maintain a consistent relative orientation. Point cloud comparison was performed using the M3C2 algorithm, with a projection cylinder radius set to 1.5 cm. The 3D surface model derived from the camera rig was used as the reference cloud. To simplify computation and align with the predominantly planar soil surface, the normal estimation

mode was set to vertical, using the Z-direction for projection.

To assess the relative accuracy of point clouds generated by the camera rig system, the PMs and their standard deviation components in the X, Y, and Z directions over the period from June 2020 to June 2021 were analyzed. Core points from the first point cloud were initially selected using the farthest point sampling algorithm (Eldar et al. 1997), ensuring spatially uniform distribution. PM values from subsequent reconstructions were interpolated onto these fixed core point locations,

enabling a consistent comparison of temporal PM variability across the time series. Temporal series of standard deviation values were obtained by linear interpolation from the NN in the subsequent point clouds of the series. To investigate the spatial character of the collected temporal variations, the core points were clustered into accuracy groups using *k-means clustering* by the Euclidean distance measures mean and standard deviation of its inherent time series.

### 3.6 Soil surface change

To evaluate the suitability of the proposed approach for monitoring soil surface changes, data from all three monitoring posts were analyzed over a measurement period of three and a half years, spanning from summer 2020 to the end of 2023. In addition to daily average change detection, a focused analysis was conducted during a period of significant rainfall on freshly tilled soil. Specifically, three 3D surface models – representing the conditions before rainfall, after light rainfall, and after heavy rainfall – were compared against a baseline 3D surface model acquired immediately after tillage, which served as a reset of

the field conditions. The 3D surface model of 2021-08-24 was selected as the reference epoch. Comparisons were then made with the surface models from 2021-08-25, 2021-08-31, and 2021-09-04, representing sequential changes in surface morphology caused by rainfall of varying intensity.

### 4 Results and discussion

The section presents the accuracy evaluation and the results of soil surface change monitoring using high-resolution 3D models

derived from the multi-camera rig system. Absolute and relative accuracies are assessed, followed by a detailed analysis of erosion and compaction processes in response to rainfall events. Elevation changes are interpreted across spatial and temporal scales, revealing clear links between surface dynamics, soil management practices, and weather conditions. System performance and data quality over the multi-year observation period are also discussed, highlighting both methodological challenges and potential improvements. Animations illustrating daily soil surface changes at all three slope positions

throughout the years are provided in the supplementary material.

## 4.1 Absolute accuracy

The comparison of the SfM point clouds from all three monitoring stations with the TLS and UAV reference point clouds revealed an average height deviation of 8-12 mm. These discrepancies are largely attributed to registration errors of the individual systems within the local coordinate system. Registration errors of the TLS point cloud may also contribute, as it could not be verified whether the point cloud was registered correctly. After adjusting the height offset by applying the estimated bias, the M3C2 distances were recalculated to emphasize the local differences (Fig. 12). The central regions of the RoI show the smallest deviations (< 5 mm) in both, UAV- and TLS-based comparisons, consistent with the internal precision of the UAV and TLS measurements. Deviations increase toward the edges of the RoI (5-15 mm).

Systematic spatial patterns in the deviation maps suggest the presence of spatially correlated errors in the camera rig data. This is evident in the positive height shift in the western part of the RoI observed in the middle rig dataset, which appears in both UAV and TLS comparisons, indicating it is not an artifact of a single reference dataset. Additional deviations arise from differences in sensor perspective relative to the soil surface. For instance, wheel tracks are particularly visible in the TLS-based comparison (differences > 20 mm). This can be explained by the oblique viewing angle of the TLS system, which increases shadowing effects and potential geometry-related errors (e.g., Eltner & Baumgart, 2015). Larger deviations in wheel tracks are also visible in the UAV data, though of smaller magnitude (≈ 10 mm), likely due to the coarser resolution resulting from the greater imaging distance compared with the camera rig.

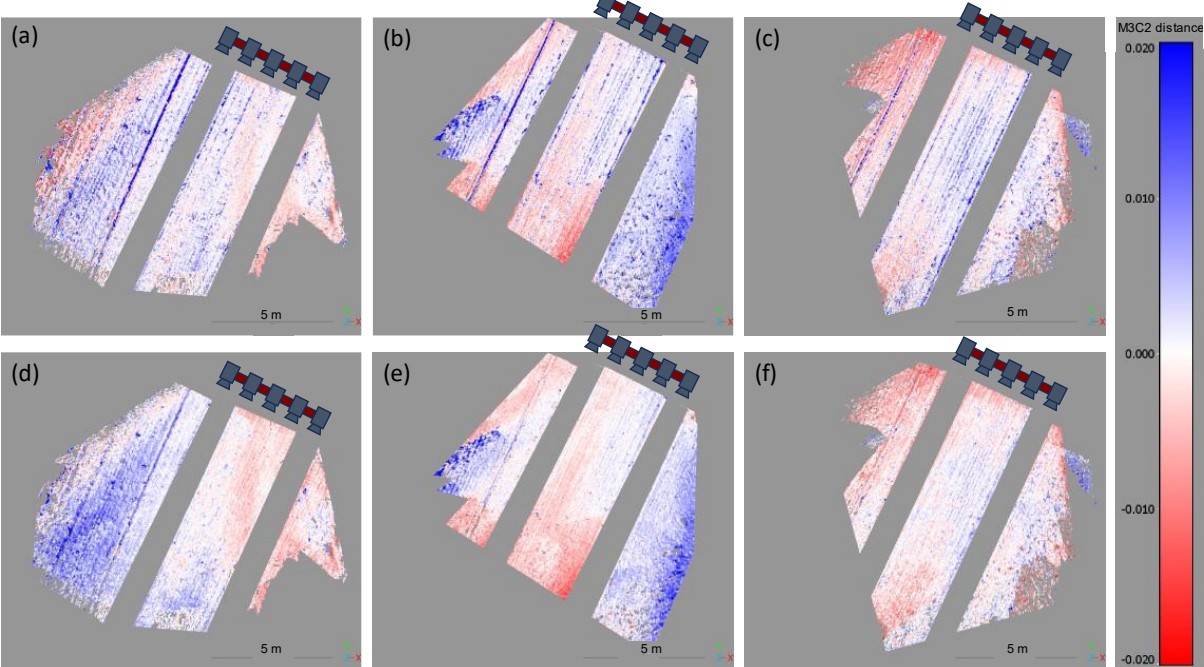

**Figure 12: M3C2 (multiscale model to model cloud comparison) distances in [m] between dense clouds of observation stations (left: top station; middle: middle station; right: bottom station) and the TLS point cloud (top row) and UAV point cloud (bottom - row), respectively. Camera positions are indicated by the camera sketches.**

## 4.2 Relative accuracy

Figure 13 shows the spatial and temporal development of the PM and corresponding LoD values at the sampled core points over a one-year period. Accuracy values are particularly high in the central part of the RoI – in proximity to the central camera – and, as expected, decrease concentrically with increasing distance. Lateral accuracy at close range is approximately 2 mm, gradually deteriorating to ~30 mm at the outer edges of the RoI. The standard deviation of the time series is less than 1 mm in the near range and increases to ~3 mm at greater distances. Correspondingly, height accuracy is about 2-3 mm in the near

range and increases to ~11 mm at larger distances. The standard deviation of the height accuracy time series is more uniform, with values of ~1 mm.

Because LoD values are derived from the PMs, they exhibit the same concentric spatial pattern. At close range, LoD values typically range between 5-8 mm, increasing beyond 10 mm at distances of ~6 m. The standard deviation of the LoD time series throughout the RoI is 1-2 mm. In terms of temporal stability, lateral accuracy can be considered stationary over the one-year observation period, with mean values consistently below 2.5 mm. However, height accuracy and LoD values increased from < 2 mm to 3 mm and from < 5 mm to 7 mm, respectively. Error fluctuations increased with distance from the camera rig, and two periods of significantly elevated errors were observed. December 2020 and April 2021.

In general, we acknowledge that the system's accuracy decreases toward the edges of the RoI. While this spatial bias could affect erosion calculations, especially when analyzing the full area, we contend that a rigid, universal threshold for restricting analyses is not feasible, as the error distribution is site-specific. Therefore, we considered the precision maps, incorporating spatially resolved 95% confidence intervals, to provide site- and event-specific LoD. Nevertheless, for the most reliable results, we recommend focusing analyses on the central region of the RoI.

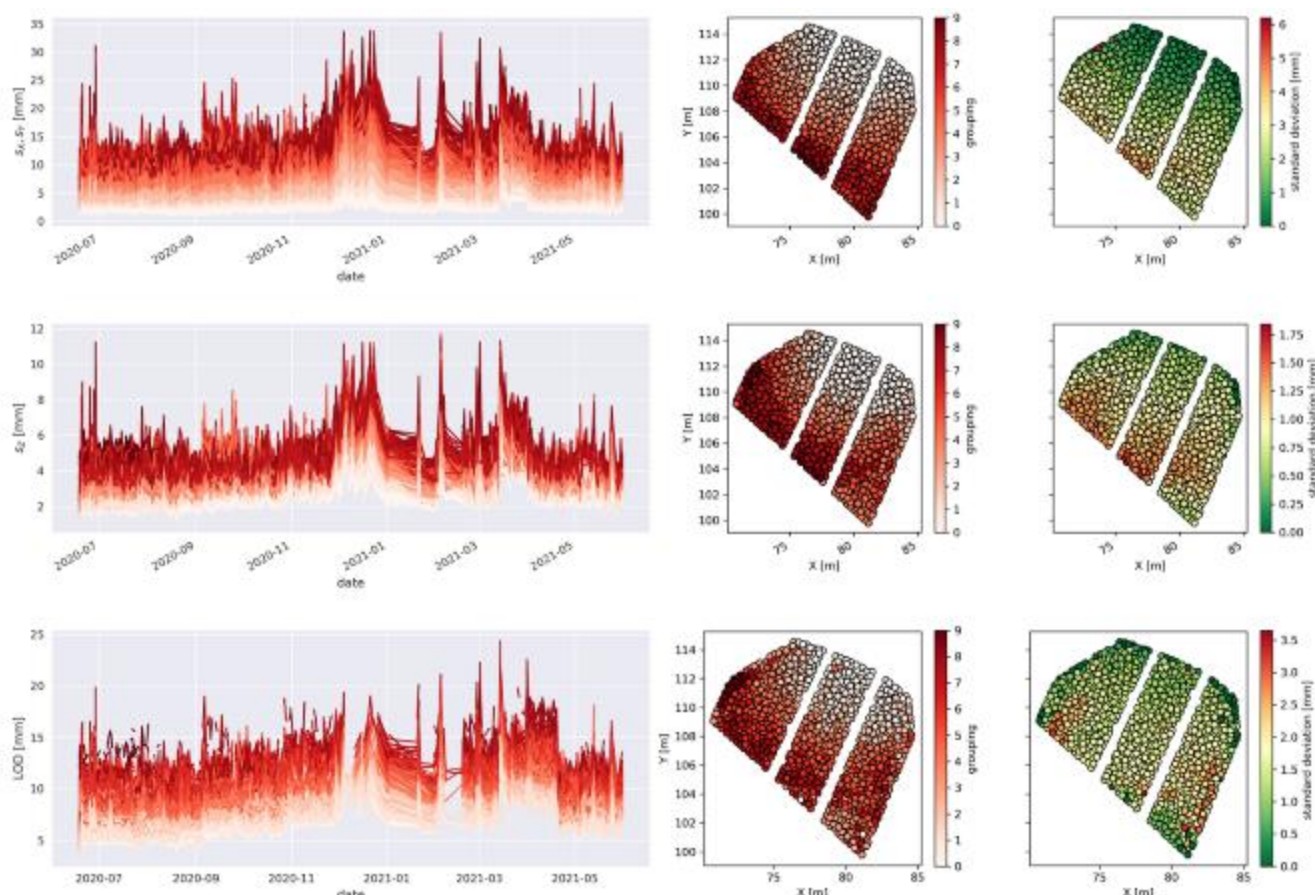

**Figure 13: Variation of lateral deviation s$_X$, s$_Y$ (top row), height deviation s$_Z$ (mid row) and LoD from M3C2-PM calculation (bottom row) on evenly distributed core points. Left-hand group of diagrams shows the development of the respective parameter over a period of one year. The middle group of diagrams shows the spatial distribution of each point, colored by its respective accuracy group label. The right-hand group shows the core points underlying time series' standard deviation around the mean of the time series.**

### 4.3 Level of detection for different events

Figure 14 shows the M3C2 distances (top row) and corresponding significant changes (middle row) for three selected epochs, each representing a different stage of soil surface change: no visible erosion, light change, and strong change.The first epoch (2021-07-25), captured one day after the reference (and four days after tillage), was recorded under dry conditions with no rainfall. The M3C2-calculated height differences range from slightly negative values to approximately +3 mm, suggesting

minimal surface change (Fig. 14a). There are some minor movements of aggregates potentially due to gravitational processes shortly after tillage, indicated by locally large positive and negative distances. Most significant changes are concentrated on the field flanks, with the right edge showing the most pronounced activity, whereas the other areas are less noticeable (Fig 14d).The second epoch (2021-07-31), following some minor rainfall of 8.7 mm, reveals more areas of local significant change larger than 6 mm in the central RoI (Fig. 14e). Additionally, an overall lowering of the whole soil surface by 3-5 mm is observed (Fig 14b). However, this change is below the minimum detectable threshold (LoD = 6 mm) and is therefore not classified as significant.The third epoch (2021-08-04), following heavy rainfall (27.8 mm) recorded on the previous day, shows pronounced erosion patterns. In contrast to the second epoch, height changes range between 10-60 mm (Fig. 14c). Notably, rill erosion patterns become clearly visible, indicating substantial surface runoff and sediment transport in response to the intense precipitation (Fig 14f).

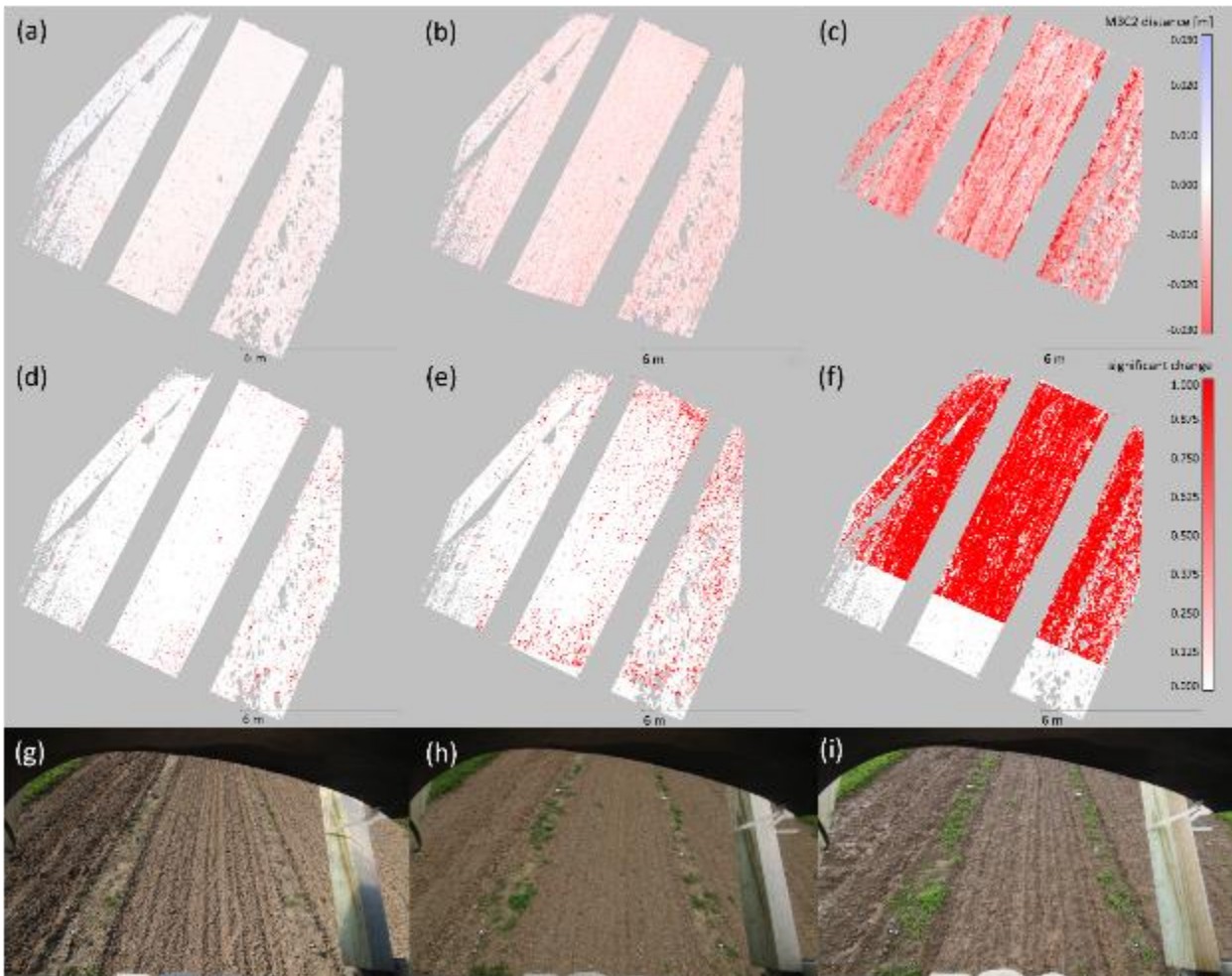

**Figure 14: M3C2 (multiscale model to model cloud comparison) distance (top row), the significant changes (middle row), and images of the area of interest (bottom row) on the days of the 2021-07-25 at the middle slope one day after reference (a,d,g – left column), 2021-07-31 after an accumulated rainfall of 8.7 mm within four days (b,e,h – center column) and on the 2021-08-04 after a heavy rainfall event with 27.8 mm (c,f,i – right column).**

### 4.4 Observing soil surface changes

Initial interpretations of the generated time series can be drawn from Fig. 15, which compares M3C2-derived elevation changes across the lower, middle, and upper slope segments alongside rainfall, temperature, and snow cover over a three-year and four-month observation period. The lower slope – characterized by a gradient of up to 14% and not yet functioning as an accumulation zone – exhibits the most pronounced elevation changes. This can be attributed to its steep inclination and the substantial upslope contributing area, which facilitates high volumes and velocities of overland flow. Sudden and substantial decreases in

elevation consistently follow periods of intense or prolonged rainfall. Notably, strong elevation responses can also be triggered by smaller rainfall events occurring within a few days after tillage (see Fig. 18 and 19). During intense rainfall events, erosion occurs primarily as interrill erosion, though rill erosion is occasionally observed at the lower slope. Major rainfall episodes in July and August 2021, for instance, led to rapid surface lowering, with average elevation losses reaching up to 1 cm.

The most significant changes tend to occur immediately after tillage, when the soil surface is loose and bulk density is low, resulting in both erosional and non-erosional processes such as compaction and consolidation (Kaiser et al., 2018, Epple et al., 2025). Even relatively minor rainfall events – such as those in October 2021 – can induce substantial elevation changes when occurring shortly after field cultivation. Therefore, future set-ups should consider integrating a sediment-collecting system, at least until settling rates can be reliably quantified using a data-driven approach (e.g., Epple et al., 2025). However, it also should be noted that sediment traps have their own limitations, such as providing only averaged changes across the slope. Moreover, the absence of sediment in a trap does not imply that measured height changes are purely non-erosional. SfM photogrammetry can detect small-scale erosion and accumulation and hence localized sediment redistributions that may not reach the sediment collector at the hillslope bottom.

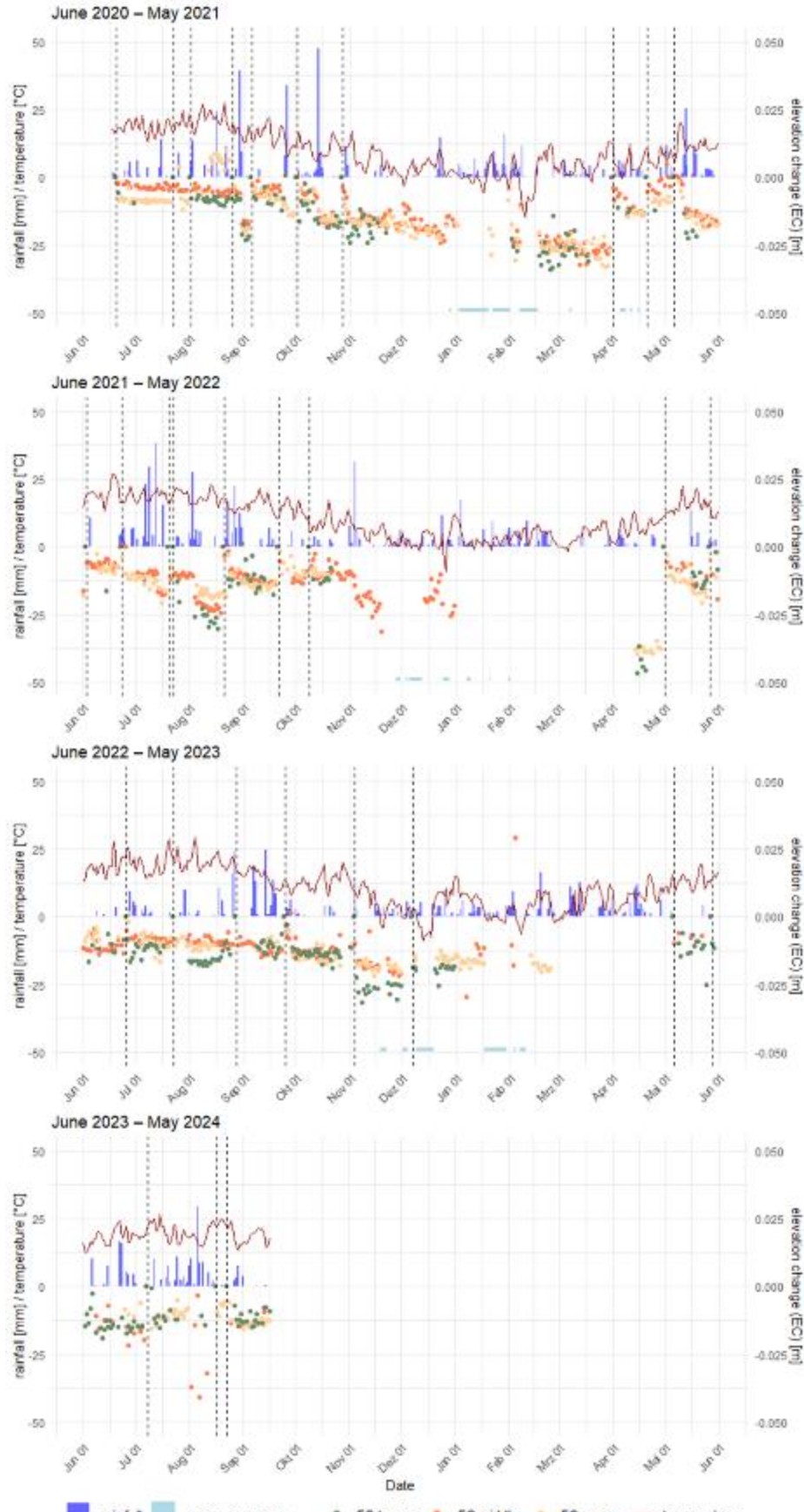

**Figure 15: Temporal comparison of the daily M3C2 height changes for the lower, middle, and upper slope position (EC lower, EC middle, EC upper, where EC = elevation change), snow coverage of the slope (covered/ not covered) and the precipitation and temperature data of the weather station (landwirtschaft.sachsen.de,station Nossen – NOS) for the period June 2020 until September 2023. Vertical dotted lines = soil tillage/ new reference day.**

465

Beyond providing insights into individual rainfall events, the time series also enable the assessment of seasonal developments, particularly during the winter months. On days with snow cover – indicated in Fig. 15 by light blue markers along the bottom – elevation change measurements are not possible, as the soil surface is obscured and image matching fails due to the lack of surface texture. Exceptions are observed on 2023-02-24 and 2021-04-08 (Fig. 16b and a), where positive elevation changes of +2.9 cm and +0.5 cm, respectively, were detected as a result of intense frost. Between November and April, a general elevation decrease ranging from 3-5 cm is evident, attributed to both late autumn rainfall events (e.g., November 2021) and snowmelt processes, as reflected in the February 2021 data.

Occasional positive elevation changes during this period can also result from factors unrelated to frost. For example, substantial vegetation growth before and after 2020-08-22, led to an elevation increase of up to +1.0 cm (Fig. 16c). On this particular day, the combined presence of vegetation and water accumulation from a recent precipitation event contributed to an average elevation rise of +1 cm. Similarly, in August 2023, dense vegetation growth on the upper slope (Fig. 16d) had a measurable impact on the overall elevation change. Even sparse vegetation can locally increase elevation (Fig. 16e), complicating the interpretation of erosion processes. Future research could more extensively explore vegetation dynamics and their interaction with surface change. An approach based on machine learning, such as the vegetation detection method proposed by Grothum et al. (2023), presents a promising solution for improved vegetation filtering. Applying such advanced methods to larger, more vegetated datasets, e.g., similar to those studied by Onnen et al. (2020), can be a valuable direction for future investigations.

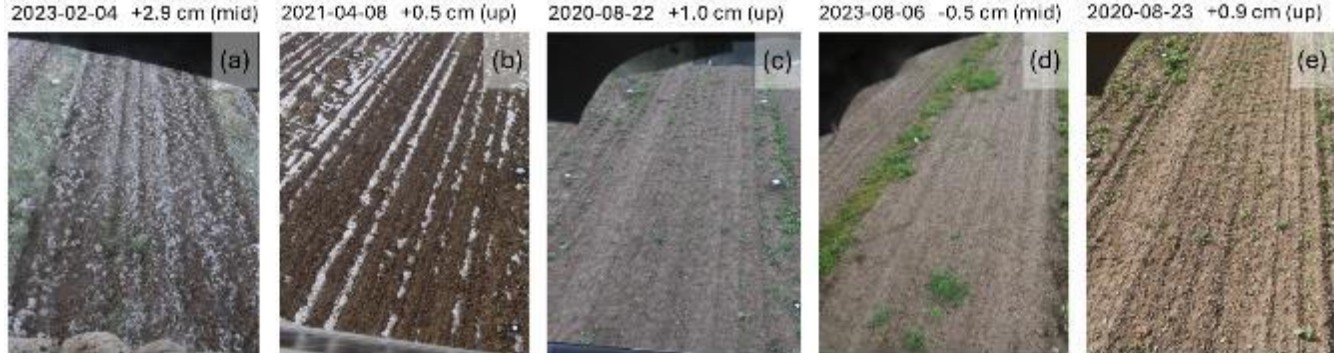

**Figure 16: Images on days of positive elevation changes/ positive development. (a) and (b) show examples of positive changes due to intense frost, in (c) during a rainfall event water covered areas arise on the surface, (d) is a combination of water coverage and plant growth of a few large plants and (e) pictures small plants all over the area of interest. The date, the elevation change in respect to the reference data, and the slope position are noted above.**

Data gaps occurred for several reasons. In November 2021, severe storm damage to the monitoring posts on both the lower and upper slopes resulted in extended data gaps for these positions (Tab. 1). Attempts to reconstruct the monitoring system were repeatedly hindered by adverse weather conditions and consequently an impassable surface for heavy machinery, delaying full functionality of camera systems until April 2022. Another system failure was recorded between 2023-02-25 and 2023-05-03. Table 1 summarizes the total number of survey days, excluding these long-term gaps. Additional daily data gaps arise when more than two cameras per post fail to capture usable compatible images, as 3D models require a minimum of three images for robust calculation.

**Table 1: Statistics of the entire survey period 2020-06-18 until 2023-09-15**

| Slope position | lower | middle | upper |
|---|---|---|---|
| **Survey period in days*** | 1174 | 1314 | 1155 |
| **Days on which 3D models could be created** [%]** | 61 | 67 | 80 |
| **Of this, days with defective 3D models*** [%]** | 9 | 9 | 14 |

| Total of usable days [%] | | 55 | 61 | 69 |
|---|---|---|---|---|

*without long-term data gaps due to e.g., storm damage*
**e.g., at least three images for 3D model generation*
***GCP movements due to e.g., animals, tillage, storm events*

Some 3D models were excluded due to defects, which may result from inaccuracies in GCPs, failed GCP detection, extensive vegetation growth, snow cover, water cover or challenging weather conditions. In regard of GCP failures, we minimized this influence by excluding instable GCPs, detected during the bundle block adjustment by the presence of significantly elevated error magnitudes. Where feasible, these points were re-measured using either a total station or an UAV for photogrammetric data collection. We had established a redundant network of GCPs, which mitigated the impact of potential point movements on the final georeferencing accuracy. Models listed as incalculable in Table 1 resulted primarily from the complete loss of GCPs, often after winter or following agricultural tillage. Any residual potential GCP movements were detected and accounted for during the processing workflow. According to these limitations, data is available for 55-69% of the total survey days, providing insights into both daily and seasonal elevation change trends. The lower measurement station showed an exceptional amount of outliers, due to irregular outage of camera systems.

Analysis of the variogram categorized by rainfall amount reveals that during the first 17 days following soil cultivation, variance remains consistently low across all precipitation classes (Fig. 17). After this period, accumulated low, medium, and high rainfall levels are associated with a noticeable increase in semivariance, with the most pronounced effects observed for medium and high precipitation intensities. As the time since tillage increases, precipitation can accumulate over several preceding days while also intense rainfall events may occur. Both scenarios contribute to more variable changes in surface elevation. The apparent decline in semivariance within the very high rainfall class beyond approximately 20 days is likely attributable to random noise rather than a genuine decrease in variability and should therefore be interpreted with caution. Additionally, the widening of confidence intervals with increasing time since the last tillage event further supports the trend of rising variability. Overall, Fig. 17 suggests a key threshold occurs around day 16 post-tillage, marking a notable shift in elevation change dynamics.

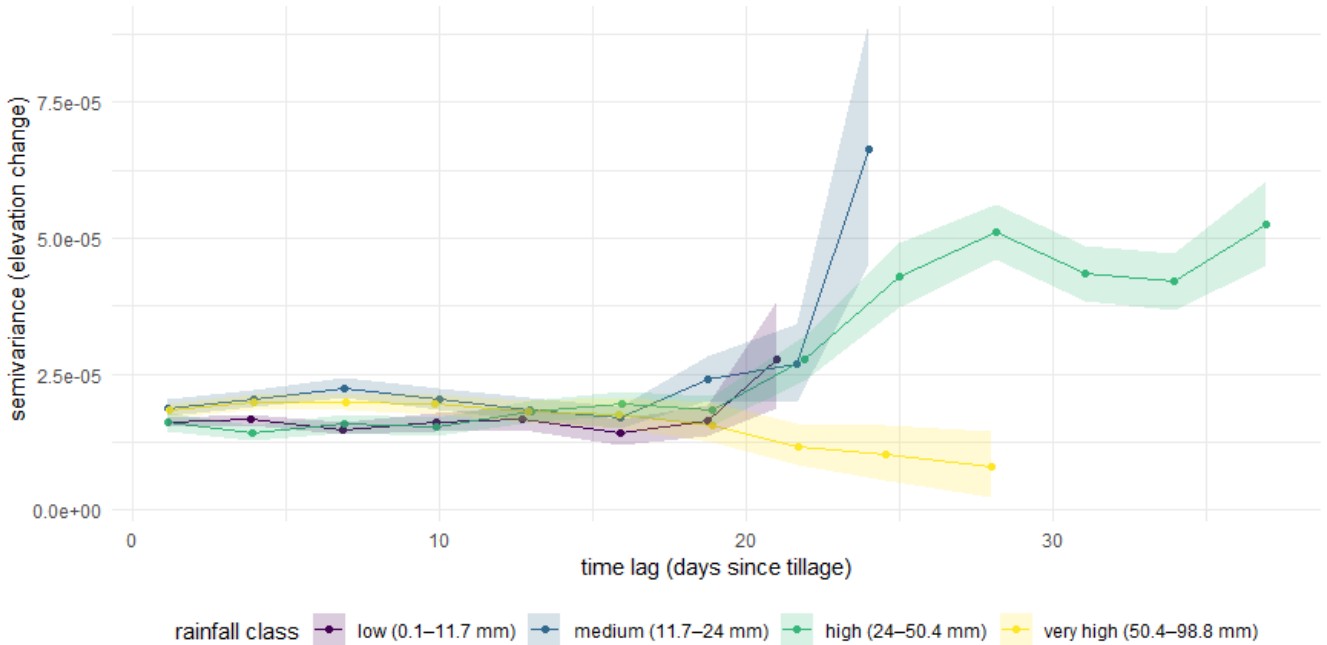

**Figure 17: Directed temporal variogram of elevation change and days since last tillage with a 95% confidence interval (lag width = 3). The data is grouped by rainfall intensity into four classes divided by 25%-quantiles.**

As shown in Fig. 18, all slope positions exhibit a negative trend: the higher the accumulated rainfall amount, the greater the accumulated elevation decrease. Contrary to expectations, no substantial differences in the strength of correlation were found

between the middle and upper slope positions. While the lower slope also indicates a negative trend, this relationship is not statistically significant. This can likely be attributed to higher measurement uncertainty and more frequent data outages at the lower slope monitoring station, which compromise the reliability of the data compared to the middle and upper slope stations. Building on the variogram analysis, different time classes were established based on the duration since the last soil management event, and correlations were examined separately for the lower, middle, and upper slope positions (Fig. 19). Given the substantial increase in variance over time (Fig. 17), particular emphasis is placed on the first two time classes. These clearly show that during the initial period (days 1-7), a strong negative correlation exists between precipitation amount and surface elevation change across all slope positions ($r < -0.5$). This negative correlation decreases during days 8-15, consistent with increasing variability and accumulated rainfall events over time.

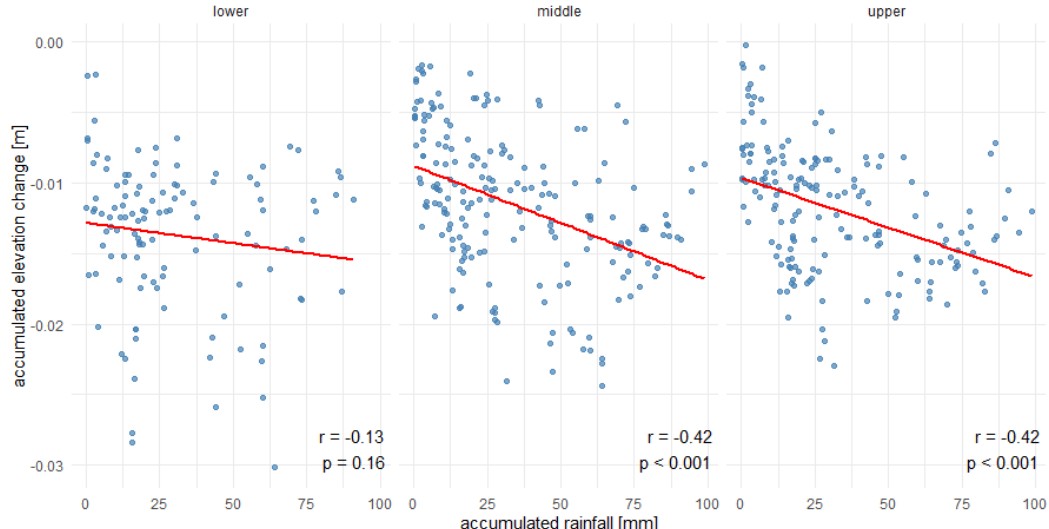

**Figure 18: Scatterplot of accumulated rainfall intensity in mm and the amount of accumulated elevation change in m for the lower, middle and upper slope positions (left to right). Pearson's correlation coefficient (r) and significance (p-value) are individually noted at the bottom. The winter months as well as days with positive changes due to water or vegetation covering the surfaces are excluded from the analysis.**

Figure 20 reinforces the pattern already observed in Fig. 15: the strongest statistically significant negative correlation ($r = -0.52$) occurs within the first seven days following soil management, gradually diminishing across the subsequent two 'days since tillage' classes. From day 30 onward, no statistically significant correlations are observed. The initially strong negative relationship is attributed to the loose, freshly tilled soil, which is highly susceptible to compaction and settling. Even minor rainfall events during this early phase can result in substantial elevation changes, leading to a new surface level that likely reflects a 'compaction baseline'. During this period, it is difficult to clearly differentiate between non-erosional processes such as compaction and settling, and sediment yield carried downslope. Epple et al. (2025) propose an empirical approach to distinguish between these processes using high-resolution elevation change data recorded at 20-second intervals on plot scales. However, their approach has not yet been tested on the broader temporal scales represented in this dataset. After this initial phase, surface elevation changes are more likely attributable to erosional processes. Small rainfall events tend to produce only minor changes, while larger events are required to generate a more pronounced elevation decrease.

To distinguish sediment yield from other elevation change processes, a collection device that measures both discharge and sediment concentration should be installed at the bottom of the slope. However, several additional aspects must be considered, as it is usually not possible to till across such field gauges. The strip of land directly adjacent to the measuring device cannot be mechanically cleared of vegetation, and repeated tillage slightly upslope reduces surface height, potentially creating a step between the experimental plot and the measuring device. Establishing a highly compacted or even sealed section between the lowest monitoring point and the measuring device could address these issues and facilitate the measurement of surface runoff.

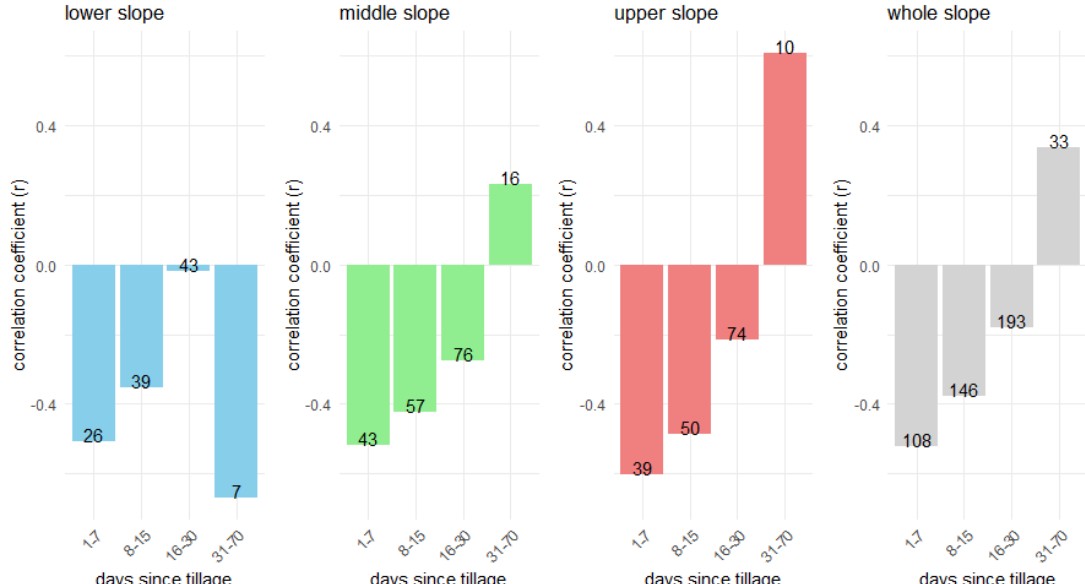

**Figure 19: Pearson's correlation (r) of rainfall intensity with elevation changes of the day after the rainfall event. Calculated for the whole slope and for all three slope positions individually. These correlations are calculated within predefined time bins of days passed since the last tillage. The count of valid observations for each bin is displayed along the bars. The winter months as well as days with changes due to water or vegetation covering the surfaces are excluded from the analysis.**

The color gradient in Fig. 20 further indicates that, during the later period (days 16-30), greater elevation changes are associated with areas that had already experienced higher accumulated rainfall in the preceding days. Compared to the first time class – dominated by a mix of erosional and non-erosional processes – these later stages require greater cumulative rainfall to produce elevation changes exceeding -0.02 m.

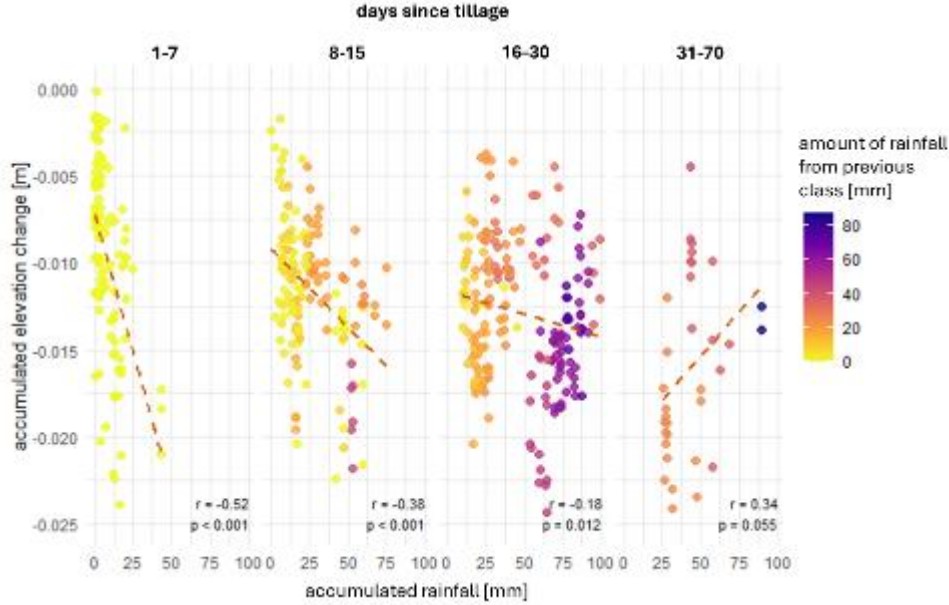

**Figure 20: Scatter plots correlating accumulated rainfall intensity in mm and accumulated elevation change in m for the whole slope, divided into four bins – days since last tillage on the slope. The data points are colored considering the amount of rainfall fallen in the previous bin. Pearson's correlation coefficient (r) and the significance (p-value) for each bin are noted at the bottom.**

## 4.5 System Evaluation and Recommendations

The time-based trigger system consistently provided daily observations of the soil surface over an extended period. The comparison of the camera rig-based 3D models with the TLS and UAV data highlights the great detail of the reconstructed soil surface. Furthermore, the reconstruction accuracy and consistency are especially high in the central area, while the errors

increase in the outer regions. This variance primarily stems from camera geometry; the central area benefits from coverage by all five cameras, whereas side areas are captured by only one side, resulting in inadequate ray intersection geometries. The PMs and LoDs reaffirm that the camera geometry significantly impacts point precision. The stationary behavior of lateral accuracy contrasts with the slight increase in height errors and LoD over time, suggesting potential changes in camera calibration, particularly in distortion parameters. Two notable data gaps with elevated values are attributed to adverse weather and surface conditions, such as prolonged snow cover (January, February and December 2021).

This study presents a novel observation system to study natural rain-driven soil surface change processes. To enhance its functionality, several improvements are suggested. The implementation of a telemetric system is proposed to relay operational status information, including malfunctions, errors, and battery levels of both the master station and cameras, to a centralized hub. This centralized hub can serve as a focal point for data management, allowing for the establishment of an integrated pipeline from data acquisition through to processing. By recognizing failing cameras at an early stage, problems can be rectified immediately, and a higher data density can be achieved. Furthermore, the wind direction should be carefully considered in regard of system orientation because during our observation period, especially during strong rainfall events, measurement quality decreased significantly or failed due to raindrops on the glass.

Currently, no single open-source software solution fully meets this study's requirements for ease of use, comprehensive code documentation, and extensive error analysis. While some options, like MicMac (Rupnik et al., 2017), are powerful, their complexity makes them less accessible to users without a sufficient background in photogrammetry. COLMAP (Schönberger & Frahm, 2016) offers an alternative, but its georeferencing capabilities are limited. A detailed comparison of available software is beyond the current scope of this study, although future work should focus more extensively on open-source solutions.

Moving forward, leveraging the generated 3D surface models to test soil erosion models against them is recommended. This comparative analyses will provide valuable insights into the applicability and efficacy of the models within the context of soil erosion modeling complexities. However, we need to acknowledge that the initial data collection had several gaps, primarily due to inexperience with the novel monitoring setup. We expect that future studies replicating our design will benefit from our lessons learned, resulting in far fewer interruptions and more complete datasets. Data continuity was largely maintained by operating three independent camera rigs simultaneously. This redundancy ensured that even when one rig experienced an issue, at least one other was actively collecting data. This multi-rig approach is a key design feature that minimizes data loss and ensures that model test efforts are not significantly compromised. The most substantial data gaps occurred during the winter months, when frost and snow cover meant that no significant changes in the soil surface were expected. These periods of inactivity do not hinder our test goals, as our primary focus is on event-based soil erosion rather than long-term, annual averages. Our methodology is specifically designed to test process-oriented, event-driven models (such as RillGrow). The high spatio-temporal resolution of our data allows us to capture the dynamic, process-level changes in soil surface during specific rainfall events. This makes our data particularly valuable for models that simulate discrete erosion processes. Conversely, our approach is not intended for the testing of empirical, annual-scale models like RUSLE, which rely on long-term average values and are less sensitive to the high-frequency changes captured by our system. Therefore, the gaps that occurred due to hardware issues or power supply disruptions, while a point of consideration, do not invalidate the dataset's utility for its intended purpose. Our data remain well-suited for its intended test purpose despite the existing gaps, particularly when focusing on the periods of active erosion events. Such model test efforts are crucial for advancing our understanding of soil erosion processes and refining predictive process-oriented modeling techniques (e.g., Eltner et al., 2025).

## 5 Conclusion

We introduce a camera-based soil surface change observation to assess natural rainfall effects. Our setup comprised five cameras mounted on three sturdy wooden structures each and activated by a rain gauge through a microcontroller, facilitating event-based and time-based triggering. We established a Python-based workflow to automate image synchronization, GCP detection, point cloud processing and comparison. The resulting processed point clouds offer a detailed overview of the observed epoch in comparison to utilizing TLS or UAV photogrammetry. Notably, they exhibit minimal systematic errors, showcasing the effectiveness of our stationary observation system in capturing surface topography with greater detail.

For a comprehensive evaluation, we conducted a year-long time series analysis, comparing each epoch with a reference epoch using M3C2-PM. The standard deviation of these PMs ranges from 3 mm laterally and 2 mm vertically in the best cases to 30 mm laterally and 20 mm vertically in the worst cases. The accuracy is influenced by the distance from the cameras and the position of points relative to the camera station, with the LoD varying between 5 mm at best and 25 mm at worst.

Our established workflow is readily transferable to similar experimental settings. The resulting unique dataset provides valuable insights into both daily and seasonal dynamics of soil elevation changes. Notably, the most significant changes occur even after relatively low rainfall events following agricultural tillage, likely due to a combination of erosional and non-erosional processes. A consistent decline in elevation is observed throughout the winter months. This data generation approach enables the analysis of erosion development at high spatial and temporal resolutions during individual rainfall events, while also capturing broader seasonal trends.

### Data and Code availability

The hardware to capture the image data, including the source code for its control, are provided in the supplement (SetupCameras). The Python scripts for the time lapse data processing are provided here: https://github.com/onlyole/TimeLapseErosion.git and in the supplement (scripts_TimeLapseErosion). The animations of the soil surface changes are provided in the supplement (animation_soilSurfaceChange). The image data and processed point clouds (of change) will be provided in a separate data publication (https://doi.org/10.5194/essd-2025-380).

### Author contribution

All authors contributed greatly to the work. OG: conceptualization, data processing, investigation, writing (original draft); LE: conceptualization, data processing, data analysis & investigation, field support, writing (original draft); AB: data acquisition, writing (review and editing); XB: data processing, field support; AE: conceptualization, methodology, figures, data acquisition, funding acquisition, writing (review and editing). All authors have read and agreed to the published version of the paper.

**ACKNOWLEDGEMENTS**

The study was supported by the German Research Foundation (DFG) under the project number 405774238: *High resolution photogrammetric methods for nested parameterization and validation of a physical based soil erosion model*. We also extend our sincere thanks to the team of the 'Versuchsstation Nossen' of the LfULG Saxony for their invaluable technical support and assistance throughout the monitoring campaign. During the revision of the manuscript, we used ChatGPT (GPT-4o) to improve the readability and language of the text. After using the tool, we carefully reviewed and edited the content as needed to ensure correctness and originality. We take full responsibility for the final content.

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
