# Peer review of "Near-continuous observation of soil surface changes at single slopes with high spatial resolution via an automated SfM photogrammetric mapping approach"

_EGUsphere, 2025_

## Author Comment (AC1)

Response: We thank both referees for their constructive feedback and the handling editor for their support. The comments will substantially improve the manuscript. Below we address each point in detail, indicating changes made or clarifications to be added in the revised version.

In this paper, a new approach for an almost-continuous monitoring of erosion at the hillslope scale is presented. The idea of conducting a continuous survey is very innovative and challenging, and the methods presented are very promising. However, the paper in its present form presents some criticisms and necessitates some revisions.

- In the "abstract", a lot of acronyms (SfM, GCP, CNN, DoD, LoD) are presented. However, most of them are not useful for the "abstract" section and could be removed.

Response: Thank you for this advice. Revised to remove acronyms that appear only once in the abstract.

- The "introduction" section is too simplistic. At least, the papers dealing with the use of SfM for monitoring hillslope erosion at the event/run scale should be presented.

Response: We will expand the introduction to include relevant studies using SfM for event- and run-scale erosion monitoring.

- Line 72: The start of the sentence is a bit twisted. Please, revise.

Response: We changed the sentence for clarity.

- Line 77: It is not clear if the investigated field was hydraulically delimited or not. This implies significant differences in terms of runoff generation and sediment transport dynamics.

Response: The system was installed so that the border of the area of interest remained unchanged and covered by vegetation, while the observed region itself was regularly tilled. This created a natural boundary between the monitored and surrounding areas. We will add this information to the manuscript.

- Line 79: I think that the "e" should be deleted.

Response: Thank you for noticing. We will remove it.

- Some acronyms are not defined in the text.

Response: We will check the manuscript carefully to avoid such issues.

- The sequence of tenses is not always optimal. Please, revise the text.

Response: We will revise the text to ensure temporal consistency.

- Lines 101-102: The sentence is quite confusing, and some words are repeated.

Response: We will revise it.

- The quality of the legends in Figure 3 is too low.

Response: In the final manuscript version high resolution figures will be used.

- Line 432: there is a point after mm that should be deleted.

Response: Thank you for noticing. We will remove it.

- How do you discern the elevation changes due to vegetation or post-tillage settlement from those due to erosion and deposition phenomena? I believe that resolving this aspect is crucial for the satisfactory application of the presented methodology. Even if briefly discussed, it remains incomplete.

Response: Thank you for your comment. We will clarify the issue of vegetation cover and its impact on the accuracy and reliability of soil erosion measurements. Regarding post-tillage settlement we already specify that we measure soil surface changes rather than soil erosion per se, but we will make this distinction even clearer in the revised manuscript.

- Have you thought about ways to increase the percentage of total usable days? Is it possible to further protect the setup and avoid big gaps in data collection?

Response: Yes, it is for sure possible to avoid such large gaps by improving the power supply and als further refining the data processing. We will discuss the in more detail in the revised manuscript. However, we would also like to highlight, even with 40% data gap, we have a lot more observation data for soil surface changes available than ever before as beforehand no such event data was available at all.

References:

Eltner, A., Kaiser, A., Abellan, A., Schindewolf, M. (2017): Time lapse structure-from-motion photogrammetry for continuous geomorphic monitoring. Earth Surface Processes and Landforms, 42(14), 2240-2253

Onnen, N., Eltner, A., Heckrath, G., Van Oost, K. (2020): Monitoring soil surface roughness under growing winter wheat with low altitude UAV sensing. Earth Surface Processes and Landforms, 45(14), 3747-3759

---

## Author Response (AR2)

Response: We thank both referees and the Pedro Batista for their constructive feedback and the handling editor for their support. The comments substantially improved the manuscript. Below we address each point in detail, indicating changes made or clarifications added in the revised version.

Topic Editor:

Thank you for responding to the referees' comments in the interactive discussion. Please revise your manuscript accordingly. In addition, please revise the paragraph in lines 577-580: it is not clear if you are recommending to "validate" the 3D surface models against soil erosion models or the other way around. The last sentence of the conclusion also needs rewriting: "future research should focus on modelling ... of soil erosion models" does not make sense to me. Overall, I would not put so much emphasis on erosion modelling (e.g. I would not mention erosion modelling at the end of your conclusion). Your monitoring approach, although certainly useful for testing erosion models, is valuable on its own. We need to improve how we measure erosion in the field, and this is a step in the right direction, despite current limitations. In any case, when it comes to modelling, I would prefer it if you use terms such as "model testing" or "model evaluation" instead of "validation". See Beven and Young (2013) for guidelines.

Thank you, Pedro for your comments. We revised the manuscript according to our comments to the referees. We further changed referring to model validation to using the terminology test(ing) according to your guidance to Beven and Young. And we removed the concluding sentence.

Referee 1:

The manuscript introduces a novel, largely automated structure-from-motion (SfM) photogrammetric system for high-resolution monitoring of soil surface change over 3.5 years on agricultural hillslopes. Synchronized DSLR cameras (triggered by rainfall events and timers) capture daily imagery, which a custom Python workflow processes: it time-synchronizes photos, applies a convolutional neural network to detect ground control points under varying conditions, and runs Agisoft Metashape SfM to reconstruct daily 3D soil-surface point clouds. From these, daily digital surface models and change-of-surface (DoD) maps are derived at millimeter-scale resolution. The method is validated against terrestrial laser scanning (TLS) and UAV photogrammetry. The data from a freshly tilled loess field demonstrates detailed topographic changes following tillage and rainfall. Overall, this approach is innovative and promising for tracking erosion dynamics at high spatial and temporal resolution.

**Major Comments**

- The manuscript is very comprehensive, but some methodological sections (e.g. the multi-step GCP tracking and neural network architecture) are very detailed. To improve readability, consider moving deep technical specifics (such as the exact CNN architecture, clustering parameters, or algorithmic steps) to an appendix or supplementary material. This would shorten the main text while still making the detailed methods available for interested readers.

Response: We shortened and streamlined the methodological sections to improve readability. However, we prefer to keep essential technical aspects in the main text to ensure that our

approach remains understandable and reproducible. Details are made concise and thereby an appendix is avoided.

- The authors claim that "the resulting high-resolution datasets are valuable for analysing erosion dynamics and validating soil erosion models." However, the study is primarily a methodological demonstration rather than an explicit evaluation of model validation. The conclusions should be tempered to reflect this. If model validation is to be a major claim, the manuscript needs to either provide analyses that link the data directly to model performance or clearly state that testing such models is outside the current scope.

Response: Thank you for highlighting this potential misunderstanding. We adapted the abstract accordingly. The corresponding sentence (lines 24-25 of the revised manuscript) has been changed: "The monitoring system and workflow are transferable, and the resulting high-resolution datasets are expected to be valuable for analyzing erosion dynamics and validating process-based soil erosion models." However, in the results and discussion we do not want to make further statements as we believe we are not promising anything that we cannot offer. Also, in the conclusion chapter, we solely state that future research should focus on the implementation of these new measurements into erosion modelling/validation (line 605 of the revised manuscript).

- A critical limitation is that the system records all surface elevation changes, including non-erosional effects (e.g. soil compaction or tillage settling). The observed strong negative correlation between rainfall and elevation loss in the first week post-tillage indicates both erosion and compaction. However, without a method to distinguish these processes, the results cannot be interpreted purely as erosion. The authors note this challenge but provide limited quantitative separation techniques. The referenced approach by Epple et al. (2025) has not been applied at this temporal scale. The authors should discuss and, if possible, implement methods to discriminate erosional changes (e.g. mass removal) from other changes. For example, integration with sediment traps or other measurements (see below) could help verify when soil material has actually been removed by erosion versus simply redistributed or compacted.

Response: This is indeed a valuable aspect, and one we would like to emphasize for future research. Unfortunately, our measurement method, which was tested with an overflow weir and a sampler at the slope foot, proved inefficient. Small events often led only to redistribution or sedimentation in front of the device, while large events overloaded it. In addition, fauna activity caused sampler failures during some events. Practical constraints also limited our set-up: the system had to be placed far enough from the lowest station to allow tractor access across the slope. This requirement resulted in slope-parallel tillage tracks at the shallow slope foot, which further promoted sedimentation in front of the device. As a result, our approach did not produce sufficiently reliable results, and we therefore excluded sediment yield measurements from the manuscript. Nevertheless, we agree that such systems can be advantageous and have added suggestions for potential measurement device designs in the discussion section, immediately following the paragraph on compaction and settlement (lines 456-461 of the revised manuscript). Future set-ups should consider integrating a sediment-collecting system, at least until settling rates can be reliably quantified using a data-driven approach (see Epple et al., 2025). However, these systems have their own limitations, such as providing only averaged changes across the

slope. Moreover, the absence of sediment in a trap does not imply that measured height changes are purely non-erosional: SfM can detect small-scale erosion and accumulation, capturing localized yields that may not reach the slope-foot trap.

- The monitoring system achieved data on only 55–69% of survey days. Hardware issues (e.g. rigs knocked over by storms) and maintenance downtime caused significant gaps. This undermines the claim of "near-continuous" monitoring. The manuscript should acknowledge this limitation more explicitly. In particular, discuss how the 31–45% data loss affects the claimed continuity, and consider suggesting improvements for hardware robustness or system redundancy. The data gaps also impact the applicability for validating models like RUSLE that rely on annual or longer-term averages; this should be discussed. For example, if many events are missed, event-based model validation is compromised.

Response: In our setup, we indeed faced the challenge of larger data gaps. However, many of these gaps were the result of inexperience and are unlikely to occur to the same extent in the future. We anticipate that researchers reproducing our setup will benefit from our lessons learned and thus experience fewer interruptions. Data continuity was ensured by operating three rigs simultaneously, so that at least one usually recorded measurements, meaning that model validation should not be compromised. Furthermore, our approach is particularly aimed to be well suited for the validation of physically based models, as we measure event-based changes with high spatial resolution. Long-term average values are not the focus of our design for future model evaluation. Other, more substantial gaps occurred during winter, when no measurable changes were expected due to frost and snow cover. The hardware itself proved robust; most issues were related to power supply. We elaborated on these aspects in the discussion of the revised manuscript (lines 662-677 of the revised manuscript).

- The system's accuracy varies spatially, with errors growing toward the edges of the region of interest (5–15 mm). Such spatial bias can systematically affect erosion calculations, especially for analyses covering the whole area. The authors should discuss this limitation, for instance by quantifying how much area is affected by larger errors and whether analysis should be restricted to a smaller central zone for reliable measurements. Additionally, providing confidence intervals or error bars (rather than just mean errors) would clarify how spatial uncertainty influences change detection.

Response: Thank you for your comment. We would like to note that the suggested accuracy analysis for quantifying areas affected by larger errors cannot be conducted in such a systematic manner, as these errors depend on the specific setup conditions. Providing numerical thresholds could be misleading, as no fixed limits exist; instead, each site requires its own evaluation. For this reason, we consider the precision map values to provide a site- and event-specific level of detection. These already incorporate spatially resolved 95% confidence intervals, ensuring that only significant changes are considered. Consequently, we do not see additional benefit in also plotting standard deviations. We further emphasize that change detection should primarily focus on the central region, where accuracy is highest. Nevertheless, we deliberately present the less favorable regions to openly discuss the possible errors. We adopted the manuscript to make these aspects clear (see section 4.1).

- GCPs were noted to move due to animals, tillage, and storms. It is unclear how these movements propagate into final surface uncertainty. The manuscript should quantify or at least discuss the impact of GCP displacement on accuracy. For example, how often did GCPs shift beyond measurement error, and how does that translate into potential vertical error in the DSMs?

Response: If any GCPs were found to have moved, they were excluded from the bundle block adjustment and, where possible, re-measured using either a total station or UAV-based photogrammetry. As we had installed more GCPs than strictly necessary for georeferencing, the potential movement of some GCPs could be managed without impacting the results. Consequently, no issues arose regarding DSM height errors. Such movements occurred irregularly and were easily detected during the BBA, as the affected points showed error magnitudes significantly higher than those of stable points. The models listed in Table 1 as not being calculable due to GCP failures were therefore mostly the result of complete GCP losses, primarily after winter or following unfavorable tillage. Any remaining potential GCP movements were detected and corrected for. We further elaborated on this in the revised manuscript (lines 250-252 of the revised manuscript).

- High-resolution temporal data are more suitable for some erosion models than others. The discussion should explicitly address which types of models can benefit. For instance, process-based models that use detailed spatial inputs (such as RillGrow or similar rill erosion models) can leverage sub-daily or event-scale data. In contrast, simple empirical models (e.g. RUSLE) operate on annual average erosion rates and cannot directly utilize such fine temporal detail. The authors should clarify that the proposed monitoring approach is especially useful for validating physically based, high-resolution models, and acknowledge the limited benefit for annual-scale models. Explicit examples (like RillGrow vs. RUSLE) would illustrate this point.

Response: We made sure to explicitly state that our system is most suited for validating physically based, high-resolution models (e.g., LISEM, RillGrow), and less applicable to empirical annual-scale models such as RUSLE. This clarification is added both in the introduction and the discussion (lines 605-612 of the revised manuscript).

- The innovative multi-camera synchronization needs clearer explanation. The temporal drift correction procedure, in particular, should be described in more detail to ensure reproducibility. It would also help to state whether camera internal calibration was performed only once before the campaign or periodically (e.g. after storms), and whether any re-calibration was necessary during the study. Details on camera maintenance and calibration will improve confidence in the results.

Response: As part of revising the method description for conciseness, we also streamlined the explanation of the synchronization. Camera calibration was performed once before the initial setup and again after reinstallation following the rig collapse. Further calibration was unnecessary, as only approximate values for the focal length and principal point were required; these parameters are re-estimated during each model calculation. Distortion parameters were estimated once and assumed to remain temporally stable.

- The use of a 3-sigma filter to remove outliers may be too aggressive for detecting subtle erosion features; the authors should justify this choice or consider less aggressive filtering. Additionally, the chosen 5 mm grid resolution for DSM interpolation should be justified: is this based on camera resolution, expected soil roughness scale, or processing considerations? Discussing why 5 mm is appropriate (and whether coarser or finer resolutions were tested) would strengthen the methods.

Response: We chose a strict filter because combining images that do not correspond correctly would result in erroneous 3D models. The 3-sigma threshold was selected as it is a common standard or rule of thumb in geodetic measurements. This threshold was particularly relevant for images captured during rainfall events, where offsets had a greater impact, while it was less critical for the daily datasets. Given that thousands of images were collected, the aggressive filtering did not lead to significant data loss. The 5 mm resolution was chosen based on the distance between the cameras and the area of interest. We aimed to achieve the highest possible resolution without introducing interpolation artifacts due to data gaps, which would have occurred at finer resolutions. Lower resolutions caused a loss of detail and were therefore not considered, as higher resolution was achievable without drawbacks. We expanded on this rationale in the revised manuscript to make our approach clear (lines 364-368 of the revised manuscript).

- Accuracy is currently given in millimeters, which is precise but not easily interpretable in terms of soil loss or erosion impact. The authors should consider converting key error metrics into more meaningful units (e.g. tonnes per hectare per year) or comparing them to typical erosion rates. Furthermore, statistical details are needed: provide confidence intervals on accuracy estimates, and separate systematic versus random errors if possible. An analysis of how accuracy depends on surface roughness or vegetation cover would also be valuable (since rougher surfaces may degrade accuracy).

Response: Thank you for your suggestion. We prefer to report height changes in millimeters, as we are using SfM photogrammetry, an established method in erosion studies where height change is a valid and commonly reported metric. Converting height changes to sediment yield using bulk density can be misleading, as bulk density can vary spatially across the area of interest and may introduce a false sense of measurement certainty. Moreover, the process-based erosion models we aim to support typically output height changes in millimeters as well. Regarding accuracy, we already provide significant values through the M3C2-PM method, which incorporates confidence intervals. Similarly, our comparison of SfM data to TLS uses M3C2, accounting for the level of detection based on 95% confidence intervals.

Concerning roughness, we consider this beyond the scope of our current study, especially since defining roughness from 3D data remains an active research topic with open questions regarding scale and metrics. Regarding vegetation, we applied filtering to remove vegetation as much as possible and chose study areas with minimal vegetation to focus on surface changes of bare slopes. A study emphasizing vegetation cover, including larger extents as done in Onnen et al. (2020), would be an interesting topic for future research.

- Where possible, compare the SfM-based erosion estimates to traditional measurement methods (e.g. sediment traps, erosion pins) to demonstrate practical utility. This could also help in distinguishing erosional vs. non-erosional volume changes.

Response: Thank you for your comment. However, we do not have such observations available for comparison. Additionally, similar comparisons have been conducted in previous, smaller-scale studies (e.g., Eltner et al., 2017).

- The study area was a bare, conventionally tilled plot, which simplified reconstruction. Real agricultural fields often have growing vegetation, which can complicate SfM. The authors mention using a machine-learning method to filter vegetation in some cases. It would strengthen the paper to discuss how the system could be adapted for vegetated conditions (for example, using spectral filters, morphological editing, or annual timing). The transferability claims should be more specific: what additional challenges or modifications would other sites (with different climate, terrain, or vegetation) require?

Response: Thank you for the suggestion. We added some more details regarding transferability considering vegetation affects to the revised manuscript (lines 479-482 of the revised manuscript).

- A valuable addition would be to suggest or demonstrate integration with other erosion measurements. For example, combining surface change data with sediment trap records or runoff measurements could help distinguish erosion from compaction. If changes in the soil surface do not correspond to transported sediment, one could infer non-erosional processes. The manuscript should at least discuss this possibility and how future work could integrate multiple datasets.

Response: Addressed in our response to comment 3.

**Minor Comments**

Line numbering should be continuous throughout for ease of review reference.

We follow the Copernicus template, which uses discontinuous numbering.

Some abbreviations are introduced without definition (e.g. RTC, IoT, LoD, M3C2). Define all acronyms at first use.

Response: All mentioned acronyms are already defined at first use; we double-checked.

The discussion of transferability would benefit from concrete guidance: for instance, recommended mounting improvements (e.g. sturdier rigs, solar power redundancy), or software alternatives (since Agisoft Metashape is proprietary, the authors might suggest open-source SfM tools for reproducibility).

Response: Thank you for the suggestions. We refined the manuscript to provide more detailed guidance where appropriate. Regarding open-source software, currently, there is no solution that fully meets our requirements, which include ease of use, thorough code documentation, extensive error analysis options, and more. While MicMac is an option, it is not straightforward

for users without photogrammetry expertise. Colmap could also be considered; however, it has limitations concerning georeferencing capabilities. Given that the scope of our study differs, we prefer not to discuss other software in detail here. Nonetheless, we acknowledge the importance of open-source solutions and plan to focus on this in future work.

**Referee 2:**

In this paper, a new approach for an almost-continuous monitoring of erosion at the hillslope scale is presented. The idea of conducting a continuous survey is very innovative and challenging, and the methods presented are very promising. However, the paper in its present form presents some criticisms and necessitates some revisions.

- In the "abstract", a lot of acronyms (SfM, GCP, CNN, DoD, LoD) are presented. However, most of them are not useful for the "abstract" section and could be removed.

Response: Thank you for this advice. We removed acronyms that appear only once in the abstract.

- The "introduction" section is too simplistic. At least, the papers dealing with the use of SfM for monitoring hillslope erosion at the event/run scale should be presented.

Response: Thank you for your comment. However, we tend to disagree because we already list specific literature on event scale measurement at hillslopes in line 42 (five examples). We however made it clearer that these studies are on hillslopes and event-scale to avoid confusion.

- Line 72: The start of the sentence is a bit twisted. Please, revise.

Response: We changed the sentence for clarity.

- Line 77: It is not clear if the investigated field was hydraulically delimited or not. This implies significant differences in terms of runoff generation and sediment transport dynamics.

Response: The system was installed so that the border of the area of interest remained unchanged and covered by vegetation, while the observed region itself was regularly tilled. This created a natural boundary between the monitored and surrounding areas. We added this information to the manuscript (lines 83-86 of the revised manuscript).

- Line 79: I think that the "e" should be deleted.

Response: Thank you for noticing. We removed it.

- Some acronyms are not defined in the text.

Response: We checked the manuscript carefully to avoid such issues.

- The sequence of tenses is not always optimal. Please, revise the text.

Response: We revised the text to ensure temporal consistency.

- Lines 101-102: The sentence is quite confusing, and some words are repeated.

Response: We revised it.

- The quality of the legends in Figure 3 is too low.

Response: In the final manuscript version high resolution figures will be used.

- Line 432: there is a point after mm that should be deleted.

Response: Thank you for noticing. We removed it.

- How do you discern the elevation changes due to vegetation or post-tillage settlement from those due to erosion and deposition phenomena? I believe that resolving this aspect is crucial for the satisfactory application of the presented methodology. Even if briefly discussed, it remains incomplete.

Response: Thank you for your comment. We clarified the issue of vegetation cover and its impact on the accuracy and reliability of soil erosion measurements (lines 479-482 of the revised manuscript. Regarding post-tillage settlement we already specify that we measure soil surface changes rather than soil erosion per se, but we made this distinction even clearer in the revised manuscript.

- Have you thought about ways to increase the percentage of total usable days? Is it possible to further protect the setup and avoid big gaps in data collection?

Response: Yes, it is for sure possible to avoid such large gaps by improving the power supply and as further refining the data processing. We discussed this in more detail in the revised manuscript (lines 595-615 of the revised manuscript). However, we would also like to highlight, even with 40% data gap, we have a lot more observation data for soil surface changes available than ever before as beforehand no such event data was available at all.

References:

Eltner, A., Kaiser, A., Abellan, A., Schindewolf, M. (2017): Time lapse structure-from-motion photogrammetry for continuous geomorphic monitoring. Earth Surface Processes and Landforms, 42(14), 2240-2253

Onnen, N., Eltner, A., Heckrath, G., Van Oost, K. (2020): Monitoring soil surface roughness under growing winter wheat with low altitude UAV sensing. Earth Surface Processes and Landforms, 45(14), 3747-3759